# UNDERSTANDING SELF-SUPERVISED PRETRAINING WITH PART-AWARE REPRESENTATION LEARNING

## ABSTRACT

In this paper, we are interested in understanding self-supervised pretraining through studying the capability that self-supervised representation pretraining methods learn part-aware representations. The study is mainly motivated by that random views, used in contrastive learning, and random masked (visible) patches, used in masked image modeling, are often about object parts.

We explain that masked image modeling is a *part-to-part* task: the masked patches of the object are hallucinated from the visible patches, and that contrastive learning is a *part-to-whole* task: the projection layer hallucinates the whole object representation from the object part representation learned from the encoder. The explanation suggests that the self-supervised pretrained encoder is required to understand the object part. We empirically compare the off-the-shelf encoders pretrained with several representative methods on object-level recognition and part-level recognition. The results show that the fully-supervised model outperforms self-supervised models for object-level recognition, and most self-supervised contrastive learning and masked image modeling methods outperform the fully-supervised method for part-level recognition. It is observed that the combination of contrastive learning and masked image modeling further improves the performance.

## 1 INTRODUCTION

Self-supervised representation pretraining has been attracting a lot of research efforts recently. The goal is to train an encoder that maps an image to a representation from visual contents without the necessity of human annotation, expecting that the encoder benefits the downstream tasks, e.g., segmentation and detection.

There are two main frameworks: contrastive learning[1] and masked image modeling. Contrastive learning aims to maximize the agreement of the embeddings of random augmented views from the same image. Masked image modeling partitions an image into masked patches and visible patches, and makes predictions for masked patches from visible patches. Figure 1 gives examples of random views for contrastive learning and masked and visible patches for masked image modeling.

We observe that a random view and a set of masked (visible) patches usually contain a portion of an object. It is also reported in self-supervised learning methods, e.g., DINO (Caron et al., 2021) and iBOT (Zhou et al., 2021), that different attention heads in ViTs can attend to different semantic regions or parts of an object. In light of this, we attempt to understand self-supervised pretraining by studying the capability that the pretrained encoder learns part representations.

We present a *part-to-whole* explanation for typical contrastive learning methods (e.g., SimCLR (Chen et al., 2020), MoCo (Chen et al., 2021), and BYOL (Grill et al., 2020)): the embedding of the whole object is hallucinated from the embedding of the part of the object contained in the random crop through a projection layer. In this way, embeddings of random crops from the same image naturally agrees with each other. Masked image modeling is a *part-to-part* process: the embeddings of the masked patches of the object (a part of the object), are hallucinated from the visible patches (the other part of the object).

---

[1] In this paper, we use contrastive learning to refer to the methods that compare random views, e.g., SimCLR, MoCo, and BYOL.

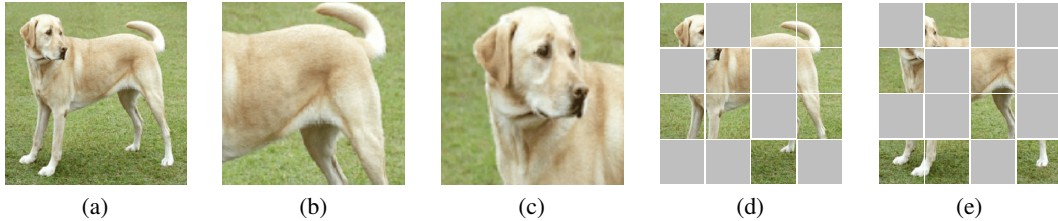

| (a) | (b) | (c) | (d) | (e) |

Figure 1: (a) original image, (b-c) two random crops, and (d-e) masked and visible patches.

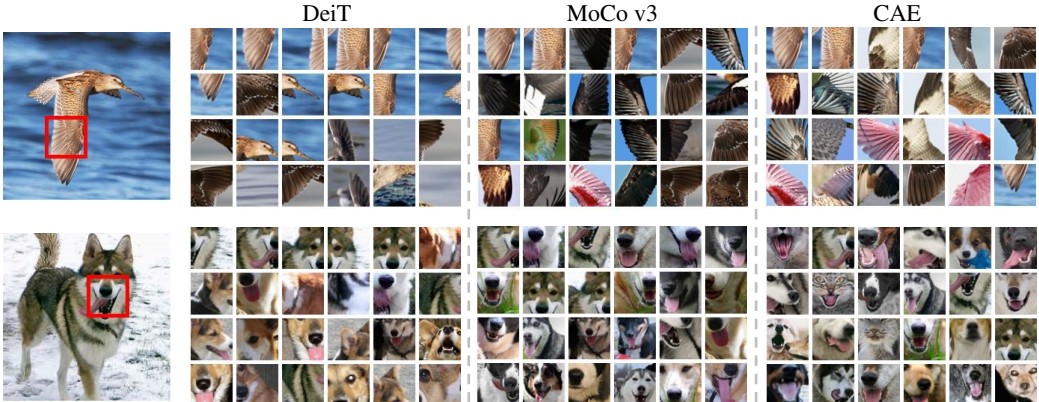

Figure 2: Top-24 patch retrieval results with three frozen encoders of DeiT, MoCo v3, and CAE, by taking the patch in the red box as the query. It can be seen that the retrieved results from CAE and MoCo v3 are about the object part (wing and dog mouth) and more precise than DeiT (about the whole object) implying that self-supervised pretraining methods, CAE and MoCo v3 are stronger at learning part-aware representations than the fully-supervised method DeiT.

We empirically compare the supervised model DeiT (Touvron et al., 2020) and typical self-supervised representation pretraining methods, including MoCo v3 (Chen et al., 2021), DINO (Caron et al., 2021), CAE (Chen et al., 2022a), MAE (He et al., 2021), BEiT (Bao et al., 2021), and iBOT (Zhou et al., 2021), on object-level recognition (image classification and object segmentation) and part-level recognition (patch retrieval, patch classification, and part segmentation). Figure 2 presents patch retrieval results using the encoders learned through CAE, MoCo v3, and DeiT, implying that the encoders pretrained by CAE and MoCo v3 are able to learn part-aware representations.

Through extensive studies and comparisons, we make the following observations. 1) DeiT outperforms contrastive learning and MIM methods except iBOT in object-level recognition tasks, which may benefit from its explicit object-level supervision. 2) In contrast, self-supervised methods learn better part-aware representations than DeiT. For example, while DeiT is superior to DINO and CAE by 0.4% and 2.3% on ADE20K object segmentation, DINO and CAE outperform DeiT by 1.6% and 1.1% on ADE20K part segmentation, respectively. 3) In contrastive learning, the encoder can learn part-aware information, while the projected representation tends to be more about the whole object. The evidence could be found in part retrieval experiments on MoCo v3, DINO, and iBOT. 4) The MIM method CAE shows good potential in part-aware representation learning. Interestingly, the method combines contrastive learning and MIM is promising, e.g., iBOT learns better representations at both object and part levels.

To summarize, this paper presents the following contributions:

- We study the capability of learning part-aware representations as a way of understanding self-supervised representation pretraining.

- We explain masked image modeling as a part-to-part task and contrastive learning as a part-to-whole task, and speculate that self-supervised pretraining has the potential for learning part-aware representations.

- We empirically compare several pretrained models on object-level and part-level recognition tasks, showing interesting findings with supporting evidence of the capability of part-aware representation learning for self-supervised learning.

## 2 RELATED WORK

**Contrastive learning.** Contrastive pretraining has been an intense academic field in the CNN era. In this work, we use it to refer to methods for comparing random views (Caron et al., 2020; Chen et al., 2020; Zbontar et al., 2021; Xie et al., 2021a; Chen et al., 2021; Caron et al., 2021), including some instance discrimination work such as (Grill et al., 2020; Chen & He, 2021; Bardes et al., 2021). As one of the representative works, SimCLR (Chen et al., 2020) learns representations through maximizing agreement between different views of the same image in the latent space. BYOL (Grill et al., 2020) uses two asymmetrical networks to bootstrap latent representation without negative samples involved during the interaction. As vision transformer (ViT) (Dosovitskiy et al., 2021) shows excellent performance via supervised learning, it is adopted subsequently in contrastive pertaining, and numerous outstanding works are proposed. For example, MoCo v3 (Chen et al., 2021) observes the hidden instability while training self-supervised ViT and solves it by using a fixed random patch projection. DINO (Caron et al., 2021) explores new properties derived from self-supervised ViT and accordingly designs a learning strategy interpreted as a form of self-distillation with no labels.

**Masked image modeling (MIM).** Masked image modeling is another self-supervised pretraining paradigm that attracts much attention recently. BEiT (Bao et al., 2021) follows masked language modeling in the natural language process (NLP) area and predicts tokens via mapping image patches by d-VAE (Ramesh et al., 2021). PeCo (Dong et al., 2021) boosts BEiT by taking into consideration more semantic information in visual tokens. MAE (He et al., 2021) learns rich hidden information by directly performing masked image reconstruction in RGB color space using ViT while SimMIM (Xie et al., 2021b) uses Swin-transformer (Liu et al., 2021). CAE (Chen et al., 2022a) adds a regressor between encoder and decoder, which is designed to align unmasked patches with masked ones, leading to a pure context encoder. Recently, a trend that combines MIM with siamese frameworks has surfaced and showed encouraging results including MST (Li et al., 2021), SplitMask (El-Nouby et al., 2021), iBOT (Zhou et al., 2021), dBOT (Liu et al., 2022), and SIM (Tao et al., 2022).

**Understanding self-supervised contrastive pretraining.** The studies on understanding contrastive pretraining (Saunshi et al., 2022; Chen et al., 2022b; Zhong et al., 2022; Wei et al., 2022) mainly focus on random augmentations (views), contrastive loss function and its variants under the assumption that: the augmentations of inputs from the same class have significant overlap in the representation space, but there is little overlap for inputs from different classes. Our work is complementary to these studies. Inspired by the observation that random views usually contain a portion of an object, and methods (Caron et al., 2021; Zhou et al., 2021) show that different attention heads in ViTs can attend to different semantic regions of an object, we investigate what the encoder and the projector do in typical self-supervised contrastive pretraining. We speculate that the pretraining task is a part-to-whole problem, predicting the representation of the whole object through the projector from the representation (obtained from the encoder) of the part of an object. We use empirical results to verify our analysis.

**Understanding self-supervised masked image modeling.** The comparison of attention in different layers between the pretrained models from MIM and the supervised approach is conducted: MIM pretraining brings locality to the trained model with sufficient diversity on the attention heads (Xie et al., 2022a). Consistent with the analysis in NLP, empirical studies are conducted in Xie et al. (2022b) to verify that MIM benefits from larger models, more data, and longer training. CAE (Chen et al., 2022a) gives the comparison between contrastive and MIM and shows MIM cares about all patches and thus achieves better results for fine-tuning. Cao et al. (2022) provides a mathematical understanding of MIM. Kong & Zhang (2022) points out that the learned occlusion invariant feature contributes to the success of MIM. In this work, we speculate that masked image modeling is a *part-to-part* process: the embeddings of the masked part of the object are hallucinated from the visible part using the position information of the masked patches, leading to better part-aware representation than the supervised model DeiT (Touvron et al., 2020).

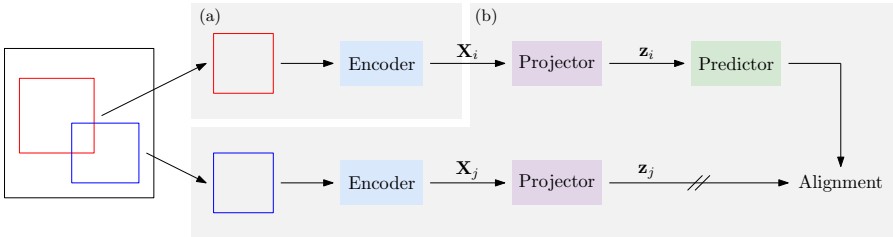

Figure 3: The pipeline of a typical contrastive learning approach. Two augmented views, red box and blue box, are generated from the original image. The augmented view in red is fed into the encoder and the projector, and then the predictor (which does not appear in earlier works like MoCo (Chen et al., 2021) and SimCLR (Chen et al., 2020)), and the view in blue is fed into the encoder and the projector. The two outputs are expected to be aligned. The gradient is stopped for the bottom stream.

## 3 UNDERSTANDING CONTRASTIVE LEARNING AND MASK IMAGE MODELING

### 3.1 CONTRASTIVE LEARNING

Contrastive learning aims to learn the encoder through maximizing the agreement between differently augmented views of the same image in the representation space. An example pipeline is depicted in Figure 3. Given an image I, the augmentations, e.g., random cropping, random color distortion, and random Gaussian blur, are applied to generate a set of $N$ augmented views, $\{V_1, V_2, \cdots, V_N\}$. An augmented view $V_n$ is fed into an encoder $\mathrm{Encoder}$, generating the encoded representation $\mathbf{x}_n$, and followed by a projector, generating the projection $\mathbf{z}_n$. The basic goal is to maximize the agreement between the projections $\{\mathbf{z}_1, \mathbf{z}_2, \cdots, \mathbf{z}_N\}$, i.e., minimize the loss

$$\mathcal{L}_{\mathrm{CPT}} = \sum_{i=1}^{N} \sum_{j=1}^{N} \ell(\mathrm{Projector}(\mathrm{Encoder}(V_i)), \mathrm{Projector}(\mathrm{Encoder}(V_j))). \tag{1}$$

In the formulation with a contrastive loss, the agreement between the projections of random augmentations from different images is minimized.

**Part-to-whole prediction explanation.** Let us consider two crops randomly sampled from the original image (see the examples given in Figure 1(b-c)). The encoded representation of the first crop is expected to describe a part of the object dog; the encoded representation of the second crop is expected to describe another part of the object dog[2]. The two representations are related but different. Contrastive learning methods project the two encoded representations into two projected representations that are expected to agree. We hypothesize that *the projection process maps the encoded part representation to the representation of the whole object*[3]. Through this way, the projected representations will agree to different views from the same image. It is assumed that the part-to-whole projection is more reliable if the encoded representation is semantically richer and is able to describe the part information. The part-to-whole process suggests that the encoder pretrained by contrastive learning methods is potentially capable of learning part-aware representations.

Figure 4 provides patch search results of a representative contrastive learning method MoCo v3 (Chen et al., 2021) based on the encoded representations before and after the projections. One can see that the results through the encoded representations are mainly about the local part, and the results through the projections tend to include the other parts of the same object. In other words, the projections tend to be about the whole object. Similar observations are also shown in Chen et al. (2022b). The search results verify the part-to-whole hypothesis.

### 3.2 MASKED IMAGE MODELING

Mask image modeling is the task of predicting some parts of an image from the remaining parts. An augmented view of an image is partitioned into patches, $\mathcal{R} = \{R_1, R_2, \ldots, R_M\}$. The task is

---

[2]In this paper, we mainly study the capability of learning representations about objects and parts, and leave the study of representations of the background as the future work.

[3]It is said that different parts have common causes in the external world (Becker & Hinton, 1992). Our hypothesis is that the common cause is the whole object.

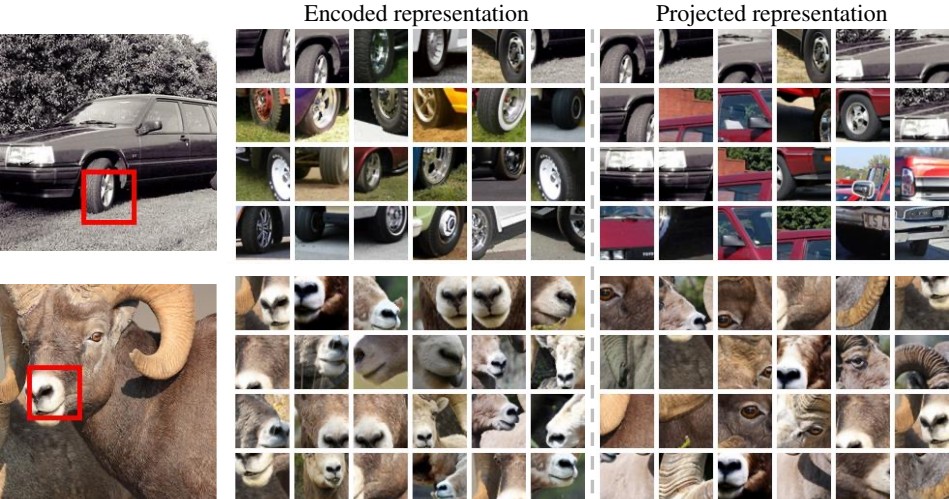

Figure 4: Illustration of patch search results using encoded representations and projections (pretrained with MoCo v3 as). Left: patch search results with encoded representations. Right: patch search results with projections. In each result, the small patch encircled by the red box is taken as the query. It can be seen that for encoded representations, the returned patches are about the same part, and for projections, the result patches are about the same object, verifying the *part-to-whole* hypothesis.

to predict a subset of patches $\mathcal{R}_m$, named masked patches, from the remaining patches $\mathcal{R}_v$, named visible patches. Considering contrastive learning that explicitly compares representations of random views, we take context autoencoder (CAE) (Chen et al., 2022a) as an example that explicitly predicts the encoded representations of the masked patches from the encoded representations of the visible patches[4].

One goal of CAE (illustrated in Figure 5), which we call masked representation modeling (MRM), is to maximize the agreement between the predictions of the representations of masked patches (through a regressor) and the representation of masked patches computed from the encoder by minimizing the loss

$$\ell_{\mathrm{MRM}}(\mathrm{Regressor}(\mathrm{Encoder}(\mathcal{P}_v)), \mathrm{Encoder}(\mathcal{P}_m)). \quad (2)$$

Here, we do not include the positional embeddings of masked and visible patches for clarity. It is noted that MRM differs from contrastive learning: MRM does not compare multiple random views, but compares the regressed representations for masked patches and the encoded representations of masked patches. In addition, there is another loss for target prediction (reconstruction) for the masked patches, which is commonly used in masked image modeling (MIM) methods:

$$\ell_{\mathrm{MIM}}(\mathrm{Decoder}(\mathrm{Regressor}(\mathrm{Encoder}(\mathcal{P}_v))), \mathrm{Target}(\mathcal{P}_m)), \quad (3)$$

where $\mathrm{Target}(\mathcal{P}_m)$ is a function to map the masked patches to the targets, e.g., d-VAE (Ramesh et al., 2021) token used in CAE and BeiT (Bao et al., 2021), or normalized RGB values used in MAE (He et al., 2021).

**Part-to-part prediction explanation.** The masked image modeling approaches, including CAE, MAE, and BEiT, make use of the positions of masked patches for making predictions for masked patches from visible patches. The visible patches and masked patches often contain different parts of an object. In other words, MIM aims to predict the masked part of an object from the visible part. We name this a *part-to-part* process. There are two part-to-part tasks: one is to reconstruct the part targets from the visible part representations (MAE and CAE) or from the visible part raw pixels (BEiT), and the other one is to regress the masked part representations (CAE). The part-to-part process suggests that the encoder pretrained by MIM methods is potentially capable of learning part-aware representations. Figure 2 illustrates the capability with the patch retrieval results.

---

[4]Some MIM methods, such as BEiT (Bao et al., 2021) and MAE (Masked Autoencoder) (He et al., 2021) do not have an explicit process to predict the encoded representations of masked patches, instead, directly reconstruct the targets.

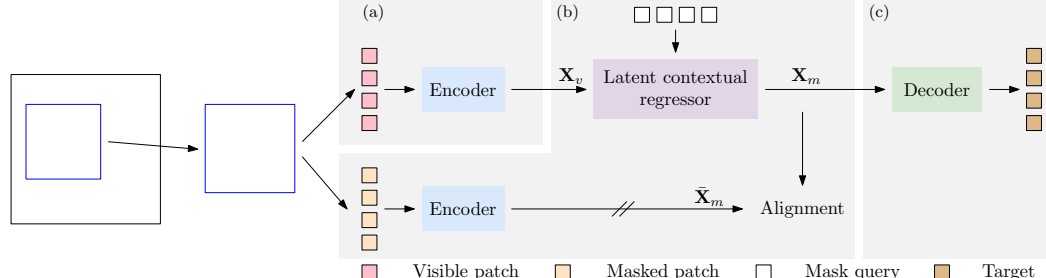

Figure 5: The pipeline of an MIM approach, context autoencoder (CAE). An augmented view (in blue) of the image is partitioned into visible and masked patches. The CAE approach feeds visible patches into the encoder and extracts their representations $\mathbf{Z}_v$ and then completes the pretext task by predicting the representations $\mathbf{Z}_m$ of the masked patches from the visible patches in the encoded representation space with latent contextual regressor and alignment constraint, and mapping predicted representations $\mathbf{Z}_m$ of masked patches to the targets. The pretrained encoder in (a) is applied to downstream tasks by simply replacing the pretext task part (b, c) with the downstream task completion part.

Table 1: Top-1 accuracy with linear probing, and attentive probing (Chen et al., 2022a), on the ImageNet classification benchmark (Deng et al., 2009).

| Method | Linear | Attentive |
|---|---|---|
| *Supervised Model*: | | |
| DeiT | 81.8 | 81.8 |
| *Contrastive Learning*: | | |
| MoCo v3 | 76.2 | 77.0 |
| DINO | 77.3 | 77.8 |
| *Masked Image Modeling (MIM)*: | | |
| BEiT | 41.8 | 51.9 |
| MAE | 67.8 | 74.2 |
| CAE | 70.4 | 77.1 |
| *Contrastive Learning + MIM*: | | |
| iBOT | 79.5 | 79.8 |

Table 2: Linear evaluation of ADE20K (Zhou et al., 2019) object-level semantic segmentation (150 classes) using $4\times$ upsampling and a single $1 \times 1$ convolutional layer on frozen backbones.

| Method | mIoU | mAcc | aAcc |
|---|---|---|---|
| *Supervised Model*: | | | |
| DeiT | 34.9 | 44.2 | 75.4 |
| *Contrastive Learning*: | | | |
| MoCo v3 | 34.7 | 43.9 | 75.9 |
| DINO | 34.5 | 43.5 | 76.1 |
| *Masked Image Modeling (MIM)*: | | | |
| BEiT | 17.8 | 23.7 | 64.9 |
| MAE | 27.1 | 34.8 | 71.6 |
| CAE | 32.6 | 42.2 | 75.2 |
| *Contrastive Learning + MIM*: | | | |
| iBOT | 38.3 | 47.4 | 78.1 |

## 4 EXPERIMENTS

We study seven representative methods with the same ViT-B encoder, including a supervised method DeiT (Touvron et al., 2020); contrastive learning methods MoCo v3 (Chen et al., 2021), DINO (Caron et al., 2021); masked image modeling (MIM) methods BEiT (Bao et al., 2021), MAE (He et al., 2021), and CAE (Chen et al., 2022a); and iBOT (Zhou et al., 2021) that combines contrastive learning and MIM. We take the training epochs specified in each work to ensure that all compared models are properly trained: 300 for DeiT, 300 (600[5]) for MoCo v3, 400 (1600[5]) for DINO and iBOT, 800 for BEiT, and 1600 for MAE and CAE. Frozen encoders are used in all experiments to understand what these different representation pretraining methods learn. More details can be found in Appendix A.1.

### 4.1 OBJECT-LEVEL RECOGNITION

We benchmark two widely-studied object-level recognition, i.e., image classification and semantic segmentation to show the capability that the pretrained encoder learns object-level representations.

---

[5]The number of effective training epochs introduced in Zhou et al. (2021).

Table 3: Part retrieval (AP, %) and classification (accuracy, %) results on the cropped part patches of CUB-200-2011 and COCO. The "Encoded" and "Projected" refer to the encoded and projected representations. "Linear" and "Attentive" columns denote the linear probing and attentive probing accuracy, respectively.

| Methods | Part Retrieval | | | | Part Classification | | | |
| | CUB-200-2011 | | COCO | | CUB-200-2011 | | COCO | |
| | Encoded | Projected | Encoded | Projected | Linear | Attentive | Linear | Attentive |
|---|---|---|---|---|---|---|---|---|
| *Supervised Model*: | | | | | | | | |
| DeiT | 35.0 | – | 44.1 | – | 90.9 | 92.9 | 88.5 | 91.4 |
| *Contrastive Learning*: | | | | | | | | |
| MoCo v3 | 50.8 | 28.4 | 52.3 | 36.8 | 93.8 | 96.0 | 92.4 | 95.3 |
| DINO | 48.9 | 31.7 | 51.8 | 41.2 | 93.2 | 95.2 | 91.7 | 94.5 |
| *Masked Image Modeling (MIM)*: | | | | | | | | |
| BEiT | 27.9 | – | 35.3 | – | 55.4 | 86.5 | 69.3 | 86.5 |
| MAE | 28.5 | – | 37.1 | – | 86.9 | 92.8 | 88.0 | 93.9 |
| CAE | 58.0 | – | 57.0 | – | 89.5 | 95.8 | 91.1 | 95.5 |
| *Contrastive Learning + MIM*: | | | | | | | | |
| iBOT | 49.3 | 31.2 | 59.2 | 41.5 | 93.8 | 95.8 | 92.1 | 95.1 |

**Image classification.** We report the linear probing, and attentive probing results of the selected models on ImageNet (Deng et al., 2009). For attentive probing, we follow the protocol in CAE (Chen et al., 2022a) that append a cross-attention layer together with a batch normalization layer and a linear classifier.

We have the following observations from Table 1. 1) The supervised model, DeiT performs better than self-supervised models at object-level recognition. 2) The models that leverage contrastive learning, i.e., MoCo, DINO, and iBOT, show superior linear probing performance than MIM-based models, demonstrating they contain more object-aware high-level semantics. 3) MIM-based models, e.g., CAE, show inferior results in linear probing while competitive results with contrastive-based methods in attentive probing. The reason might be that MIM is capable of attending to all the regions, including non-object regions in an image, thus needs a spatial feature selection step to attend to the object part, which is pointed out in Chen et al. (2022a). BEiT and MAE perform inferior, implying that the two methods are less capable of learning semantics.

**Object-level semantic segmentation.** We perform linear evaluation on ADE20K (Zhou et al., 2019) to show the object-level semantic capabilities of the pretrained models. A $4\times$ bilinear interpolation and a single $1 \times 1$ convolutional layer for pixel labeling are attached to the frozen encoder.

We can see from Table 2 that the supervised model DeiT outperforms all self-supervised models except iBOT, including contrastive learning and MIM methods on ADE20K object-level segmentation. This implies that in general the self-supervised models are not strong at object-level understanding, which is consistent with the observations for image classification. iBOT (Zhou et al., 2021), as a combination of contrastive learning and MIM, shows surprisingly better performance than the supervised model DeiT on ADE20K, implying the power of combining contrastive learning and masked image modeling for downstream tasks.

## 4.2 Part-Level Recognition

Self-supervised methods like iBOT (Zhou et al., 2021) and DINO (Caron et al., 2021) qualitatively show that different attention heads in ViTs can attend to different semantic regions of an object. We conduct the quantitative evaluation for part-aware representation obtained by pretrained models that is not well explored before, through three part-level recognition tasks, part retrieval, part classification, and part segmentation.

**Part retrieval.** We conduct part retrieval experiments on two datasets, CUB-200-2011 (Wah et al., 2011) and COCO (Lin et al., 2014). We build the part patch databases by cropping the patches centered at the keypoint. We consider four and three keypoints from the two datasets, respectively. For each keypoint, we find the minimum L2 distance ($d$) from the distances between it and all the

Table 4: Part-level linear semantic segmentation results on ADE20K-Part, Pascal-Part, and LIP datasets.

| Methods | ADE20K-Part 209 Part Classes | | | Pascal-Part 193 Part Classes | | | LIP 19 Part Classes | | |
|---|---|---|---|---|---|---|---|---|---|
| | mIoU | mAcc | aAcc | mIoU | mAcc | aAcc | mIoU | mAcc | aAcc |
| *Supervised Model*: | | | | | | | | | |
| DeiT | 27.3 | 34.7 | 69.2 | 27.4 | 36.1 | 65.8 | 41.4 | 52.6 | 73.5 |
| *Contrastive Learning*: | | | | | | | | | |
| MoCo v3 | 27.1 | 34.7 | 70.1 | 27.1 | 35.8 | 66.0 | 41.9 | 53.0 | 74.5 |
| DINO | 28.9 | 36.8 | 70.3 | 27.8 | 36.5 | 66.4 | 41.0 | 51.9 | 74.0 |
| *Masked Image Modeling (MIM)*: | | | | | | | | | |
| BEiT | 18.6 | 25.8 | 58.2 | 14.8 | 21.4 | 47.0 | 27.2 | 36.5 | 60.1 |
| MAE | 26.3 | 35.0 | 67.3 | 24.3 | 32.9 | 61.5 | 38.2 | 48.7 | 71.3 |
| CAE | 28.4 | 36.9 | 71.1 | 27.8 | 37.0 | 66.3 | 43.7 | 55.1 | 75.9 |
| *Contrastive Learning + MIM*: | | | | | | | | | |
| iBOT | 32.2 | 40.0 | 73.4 | 30.7 | 40.0 | 69.7 | 44.6 | 55.7 | 76.6 |

other keypoints in the same image, then crop a $d \times d$ patch centered at this keypoint and resize it to $224 \times 224$. We use the cosine distance as the patch distance and evaluate the retrieval performance using average precision (AP) as the retrieval metric.

The results are provided in Table 3. We have the following observations. 1) Self-supervised models except BEiT and MAE outperform the supervised model DeiT, indicating the capability that contrastive learning and CAE learn part-aware representations. BEiT and MAE perform inferior, consistent to the observations in ImageNet classification in Table 1. 2) iBOT performs the best, and the reason might be that the capability of learning part-aware representations is boosted by making use of both contrastive learning and masked image modeling.

We also report the part retrieval performance of the projected representations of contrastive learning methods in Table 3. The performance is much lower than the encoded representations. This provides an extra evidence for the part-to-whole hypothesis of contrastive learning: the projected representations are more about the whole object.

**Part classification.** We further conduct part classification experiments on the datasets used for part retrieval. We consider two kinds of extra learnable layers, linear probing and attentive probing, for classification. The results in Table 3 show that: 1) While DeiT performs the best in the image classification task (see Table 1), for part classification, contrastive-based methods like MoCo v3, DINO, and iBOT outperform DeiT by more than 2% under both linear and attentive probing settings. 2) Though MIM-based models CAE and MAE are inferior to DeiT in object-level classification (e.g., more than 10% and 4% lower in linear and attentive probing), they show competitive performance in linear probing and higher results than DeiT in attentive probing, demonstrating they learn better part-aware representations. 3) BEiT is inferior to other works, and iBOT has good performance, implying that the probing quality of pretrained encoders is a good indicator for downstream performance.

**Part segmentation.** We perform part-level linear semantic segmentation to study the finer-grained part representation modeling capability of different pretraining paradigms on three widely used datasets: ADE20K-Part (Zhou et al., 2019) containing 209 parts from the ADE20K dataset (Zhou et al., 2019), Pascal-Part (Chen et al., 2014) including 193 part categories, and LIP (Gong et al., 2017) consisting of 19 semantic human part labels. Similar to the object-level semantic segmentation experiments, linear evaluation is employed here. We maintain the same training protocols for all methods for fair comparisons. See Appendix A.2 and A.4 for dataset and training details.

The results are reported in Table 4 with the following observations. 1) Contrastive learning models, i.e., MoCo v3 and DINO, achieve competitive performance with the supervised model DeiT: DINO outperforms DeiT on ADE20K-Part and Pascal-Part, and MoCo v3 outperforms DeiT on LIP. 2) The MIM model CAE, outperforms DeiT by large margins on all three datasets, e.g., 1.1% on ADE20K-Part and 2.3% on LIP, indicating CAE learns good part-aware representations. Similar to part retrieval, possibly due to pretraining quality in representation encoding, BEiT and MAE perform

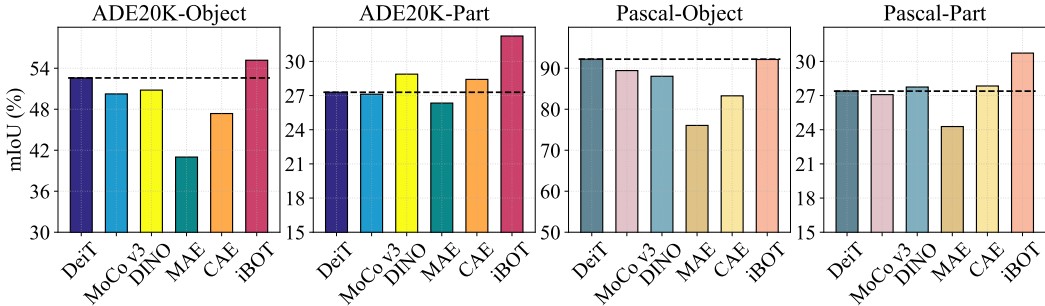

Figure 6: Comparisons between object-level and part-level semantic segmentation on ADE20K and Pascal-Part datasets. Though the supervised DeiT is superior over self-supervised models (i.e., MoCo v3, DINO, MAE, CAE) on object-level segmentation, it is generally inferior to self-supervised models on part segmentation, demonstrating self-supervised methods learn good part-aware representations. iBOT enjoys the benefits of contrastive learning and MIM. See Appendix A.3 for detailed results.

inferior. 3) Compared with object-level segmentation results in Table 2, DeiT learns better object-level semantics by explicit supervision than both contrastive learning and MIM, however, it is generally inferior to self-supervised models on part segmentation. 4) The model iBOT, which leverages both contrastive learning and MIM, outperforms all other works on three datasets, demonstrating its powerful capability in learning finer part-level semantics. Combining the two self-supervised learning techniques is thus a promising direction.

In summary, we show that self-supervised methods are potentially capable of learning part-aware representations. Among them, CAE is a representative MIM work, showing good performance by explicitly predicting the encoded representations of the masked patches in the encoding space; contrastive learning methods MoCo v3 and DINO outperform BEiT and MAE; and iBOT performs the best by combining contrastive learning and MIM. The observations are evidenced by three part-based segmentation benchmarks consistently.

### 4.3 OBSERVATION SUMMARY BETWEEN OBJECT-LEVEL AND PART-LEVEL SEGMENTATION

We conduct both object-level and part-level linear semantic segmentation on different hierarchies of the same dataset. Considering that the 209 classes in ADE20K-Part are basically chosen from 59 object classes, we denote the 59-object dataset as ADE20K-Object. Similarly, Pascal-Object consists of 16 object categories, corresponding to the 193 part categories in Pascal-Part.

The results in Figure 6 show that: although the supervised DeiT is superior over contrastive learning and masked image modeling methods on ADE20K-Object and Pascal-Object except iBOT, it is generally inferior to self-supervised models on ADE20K-Part and Pascal-Part, demonstrating self-supervised methods can learn good part-aware representations. Similar observations could be found from the object classification in Table 1 and part classification in Table 3.

In comparison to contrastive learning, CAE shows a stronger capability of learning part-aware representations, and a weaker capability of learning object-level semantics. The superiority of iBOT, a combination of contrastive learning and masked image modeling, demonstrates that it enjoys the benefits of contrastive learning and masked image modeling.

### 5 CONCLUSION

We attempt to study the capability of learning part-aware representations of self-supervised representation pretraining methods. We provide speculations for contrastive learning and masked image modeling: part-to-whole and part-to-part, with empirical results justifying the speculations. Our study presents an aspect to understand what self-supervised representation pretraining methods learn.

**Future work.** The strong capability of part-aware representation learning is one of the properties of self-supervised pretraining. There should be other characteristics that are leaved as the future work.

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

# A APPENDIX

## A.1 MODEL DESCRIPTION

For all the models involved in the experiments including DeiT (Touvron et al., 2020), MoCo v3 (Chen et al., 2021), DINO (Caron et al., 2021), BEiT (Bao et al., 2021), MAE (He et al., 2021), CAE (Chen et al., 2022a), and iBOT (Zhou et al., 2021), we use their official code to implement the encoders. It is worth noticing that for DINO and iBOT, we choose the checkpoint of the teacher models as they have been reported to perform better than the student models in their papers (Caron et al., 2021; Zhou et al., 2021).

## A.2 DATASETS

**ADE20K** (Zhou et al., 2019) is one of the most challenging benchmarks, containing 150 fine-grained semantic concepts and a variety of scenes with 1,038 image-level labels. There are 20,210 images in the training set and 2,000 images in the validation set. We choose 59 out of total 150 semantic concepts that are concrete objects containing parts (Zhou et al., 2019), termed ADE20K-Object. We also select 209 part categories that emerge both in the training set and the validation set, called ADE20K-Part.

**Pascal-Part** (Chen et al., 2014) is a set of additional annotations for PASCAL VOC 2010 (Everingham et al., 2010), thereby holding the same statistics as those of PASCAL VOC 2010. It provides segmentation masks for each part of objects. Concretely, the dataset includes 20 object-level categories and 193 part-level categories. In our experiments, we remove 4 object categories that do not contain parts including boat, table, chair, and sofa.

**LIP** (Gong et al., 2017) is a large-scale benchmark for human parsing research, which includes 50,462 images with pixel-wise annotations on 19 semantic part labels. In detail, it includes 19,081 full-body images, 13,672 upper-body images, 403 lower-body images, 3,386 head-missed images, 2,778 back-view images and 21,028 images with occlusions. There are 30,462 images in the training set and 10,000 images in the validation set. The rest 10,000 images are served as the test set with missing labels for competition evaluation.

**CUB-200-2011** (Wah et al., 2011) is a popular benchmark for fine-grained image classification, and also provides bounding box and part location annotations. It contains 11,788 images of 200 bird species and 15 part keypoint annotations per bird. In this work, we mainly leverage its part keypoint annotations. And only 4 part categories (right eye, right leg, left wing, and tail) are chosen to be considered in our experiments, to make sure that the selected keypoints are far enough away from each other and enough context information can be contained in the cropped patches. (We also tried using all keypoints and the conclusion is consistent.)

**COCO** (Caesar et al., 2018), as one of the most widely-used human pose estimation datasets, contains more than 200,000 images and 250,000 labeled person instances. Similar to CUB-200-2011 mentioned above, only 3 (nose, right wrist, and left ankle) of its 17 keypoint categories are considered in our experiments.

## A.3 DETAILED RESULTS FOR OBJECT-LEVEL AND PART-LEVEL SEGMENTATION

In this section, we provide detailed comparisons between object-level and part-level semantic segmentation in Table 5 and Table 6. Similar observations as in Figure 6 in the main paper are found: although the supervised DeiT is superior over self-supervised methods on ADE20K-Object and Pascal-Object except iBOT, it is generally inferior to self-supervised models on ADE20K-Part and Pascal-Part, demonstrating self-supervised methods can learn good part-aware representations. BEiT and MAE perform inferior, perhaps because the two methods do not have an explicit process to predict the encoded representations of masked patches, instead, directly reconstruct the targets.

## A.4 EXPERIMENT DETAILS

**Part retrieval.** In our part retrieval experiments, we directly use the pretrained encoders to extract features, without additional training process. For each method, we take the better one from the class

Table 5: Linear semantic segmentation results on ADE20K-Object and ADE20K-Part.

| Methods | Object Seg on ADE20K-Object 59 Object Classes | | | Part Seg on ADE20K-Part 209 Part Classes | | |
|---|---|---|---|---|---|---|
| | mIoU | mAcc | aAcc | mIoU | mAcc | aAcc |
| *Supervised Model*: | | | | | | |
| DeiT | 52.6 | 62.9 | 83.8 | 27.3 | 34.7 | 69.2 |
| *Contrastive Learning*: | | | | | | |
| MoCo v3 | 50.2 | 60.4 | 83.6 | 27.1 | 34.7 | 70.1 |
| DINO | 50.8 | 60.8 | 83.9 | 28.9 | 36.8 | 70.3 |
| *Masked Image Modeling (MIM)*: | | | | | | |
| BEiT | 28.6 | 37.2 | 73.4 | 18.6 | 25.8 | 58.2 |
| MAE | 41.0 | 50.6 | 79.9 | 26.3 | 35.0 | 67.3 |
| CAE | 47.4 | 58.4 | 82.9 | 28.4 | 36.9 | 71.1 |
| *Contrastive Learning + MIM*: | | | | | | |
| iBOT | 55.2 | 65.1 | 85.6 | 32.2 | 40.0 | 73.4 |

Table 6: Linear semantic segmentation results on Pascal-Object and Pascal-Part.

| Methods | Object Seg on Pascal-Object 16 Object Classes | | | Part Seg on Pascal-Part 193 Part Classes | | |
|---|---|---|---|---|---|---|
| | mIoU | mAcc | aAcc | mIoU | mAcc | aAcc |
| *Supervised Model*: | | | | | | |
| DeiT | 92.2 | 95.3 | 96.8 | 27.4 | 36.2 | 65.8 |
| *Contrastive Learning*: | | | | | | |
| MoCo v3 | 89.4 | 93.7 | 95.7 | 27.1 | 35.8 | 66.0 |
| DINO | 88.0 | 92.7 | 95.3 | 27.8 | 36.5 | 66.4 |
| *Masked Image Modeling (MIM)*: | | | | | | |
| BEiT | 56.4 | 69.0 | 76.8 | 14.8 | 21.4 | 47.0 |
| MAE | 76.1 | 84.6 | 89.5 | 24.3 | 32.9 | 61.5 |
| CAE | 83.3 | 89.7 | 93.2 | 27.8 | 37.0 | 66.3 |
| *Contrastive Learning + MIM*: | | | | | | |
| iBOT | 92.1 | 95.3 | 97.1 | 30.7 | 40.0 | 69.7 |

token or the average embedding of all patch tokens as the extracted representation. With each patch as the query patch, we calculate the cosine similarity between its representation and all the other patches' in the dataset and utilize the average precision (AP) as the retrieval metric. Finally, we average all the obtained AP scores (with all patches respectively taken as the query patch for retrieval) as the final retrieval score of the method.

Apart from the part retrieval experiments shown in Table 3, the visualized patch retrieval results in Figures 2 and 4 are obtained based on ImageNet (Deng et al., 2009) validation set. Concretely, from each pre-processed 224×224 validation image in ImageNet, we uniformly crop 49 patches sized 56×56 using a stride of 28. With all the cropped patches from the validation set, we select one patch as a query and find top 24 patches with the highest cosine similarity with it.

**Part classification.** For linear probing, we learn a supervised linear classification layer on the extracted class token of the frozen encoders. While for attentive probing, following Chen et al. (2022a), a cross attention module and a batch normalization layer without affine transformation are additionally inserted between the encoder and the linear classifier. And a new learnable class token is taken as the query of the cross attention module, to replace the original class token extracted by the frozen encoder. We use SGD optimizer with a learning rate of 0.4 and 0.04 for linear probing and attentive probing, respectively. For both linear probing and attentive probing, the models are trained for 90 epochs. And the momentum of SGD is set to 0.9, the weight decay is set to 0, and the batch size is set to 1024.

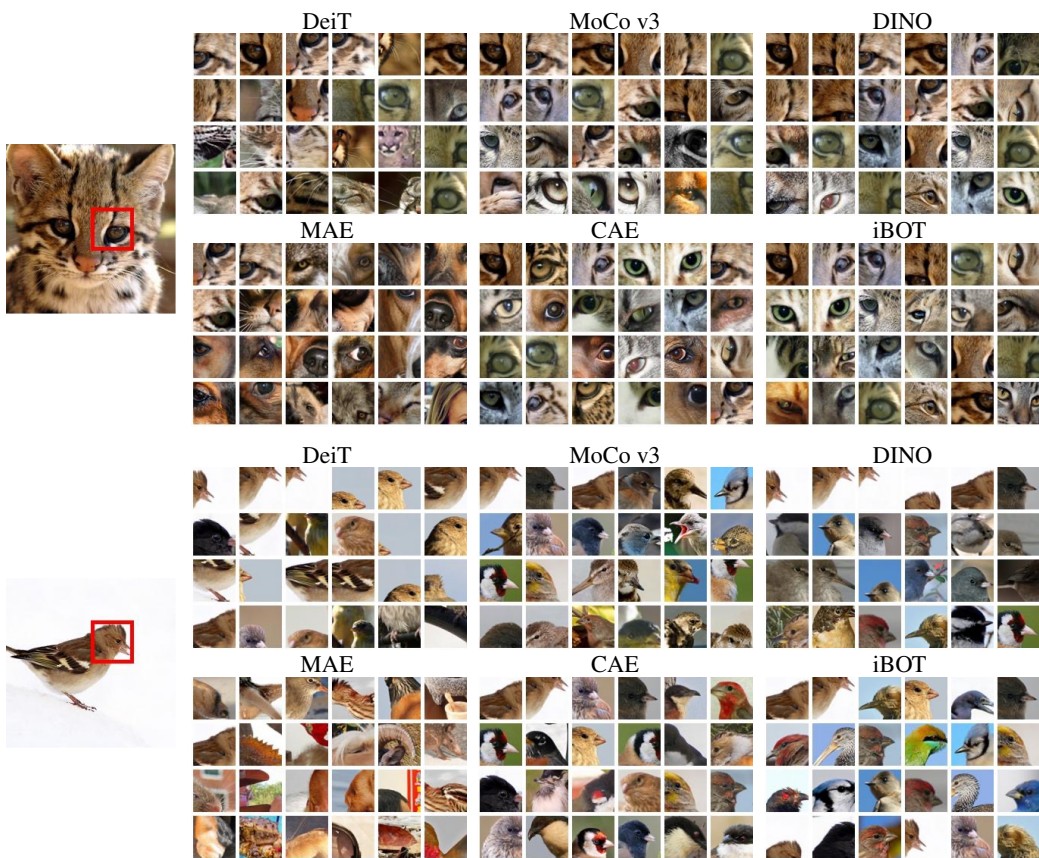

Figure 7: Patch retrieval comparisons of encoded representations on cropped patches from ImageNet.

**Segmentation.** We use the same model structure that contains a parameter-fixed pretrained encoder (e.g., MAE and DeiT) and a simple learnable $1 \times 1$ convolutional layer for object-level and part-level segmentation tasks. Note that the learning rate ($4e - 4$), training iterations ($160k$), and batch size (16) among all the experiments maintain the same during training for fair comparisons. For ADE20K, the input size is set to $512 \times 512$ following previous works (Bao et al., 2021; He et al., 2021; Chen et al., 2022a; Zhou et al., 2021). For Pascal-Part, we adopt $480 \times 480$ as image input resolution following Contributors (2020). As for LIP, we use the same input size ($320 \times 320$) proposed in LIP (Gong et al., 2017).

## A.5 IMAGENET PATCH RETRIEVAL VISUALIZATION

We visualize more patch retrieval results of the encoded representations on the ImageNet validation set in Figures 7 and 8. It is observed that the retrieved patches of self-supervised methods are generally more about the semantics of the query part than that of DeiT. The results demonstrate that the encoded representations of DeiT focus more on object-level semantics, while the encoded representations of these self-supervised methods are more about part-level semantics. Among these methods, the retrieved patches of MAE have less semantic correlation but often share similar hues.

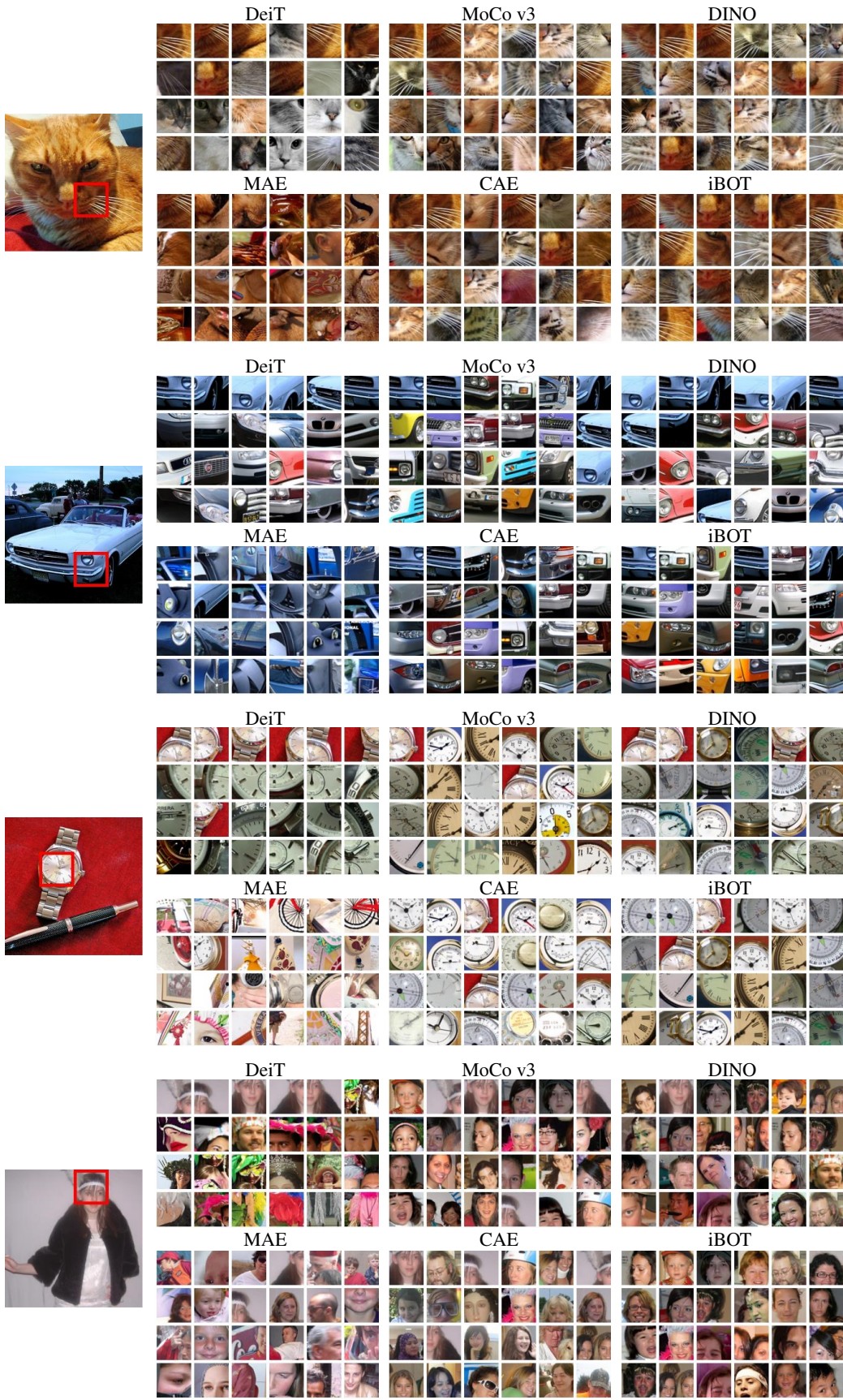

Figure 8: Patch retrieval comparisons of encoded representations on cropped patches from ImageNet.

