# OpenReview forum: "Understanding Self-Supervised Pretraining with Part-Aware Representation Learning"
_ICLR.cc/2023/Conference — Submitted to ICLR 2023_

### Official Review · Reviewer_AQH3 · 2022-10-21

**Confidence:** 3
**Correctness:** 3
**Technical Novelty And Significance:** 3
**Empirical Novelty And Significance:** 3
**Recommendation:** 6

**Clarity, Quality, Novelty And Reproducibility:**

The paper is overall well-written and clearly presented. There are some minor mistakes, in Equations (2) and (3), $P_v$ and $P_m$ are not defined. It sheds some new insights into the current self-supervised learning, which is beneficial to the community. It has a good quality in general, and the experiment details demonstrate its reproducibility. There are some claims in the paper that are not well supported from my point of view, which fails to explain some phenomena in a convincing way.

**Strength And Weaknesses:**

Strengths:
1. This paper makes an innovative claim on the current self-supervised learning methods, its conclusion leads people to a more insightful understanding of self-supervised learning.
2. Authors conduct a comprehensive empirical study on both object-level and part-level tasks to support their claims.
3. Many qualitative analyses are made, which give a clear and intuitive explanation.

Weaknesses:
1. For the part segmentation part, the authors' explanation, 'possibly due to pretraining quality in representation encoding, BEiT and MAE perform inferior', is not convincing. I do not think it is safe to draw the conclusion based on only one MIM method CAE. Therefore, whether MIM method can be simply described as a part-to-part process is questionable from the experimental results.
2. I do not see a clear reason why iBOT performs the best from the authors' explanation. It's unclear why combining part-to-part and part-to-whole processes could generate a better model. I'm afraid that the differences among the models/objectives would matter more than the differences in the training paradigm (contrastive/MIM). This could make the claims in this paper less useful for guidance on future SSL models.

**Summary Of The Paper:**

This paper tries to understand the two mainstream self-supervised methods. It speculates that masked image modeling is a part-to-part process and contrastive learning is a part-to-whole process. Both empirical results and qualitative analysis are provided to support the claims.

**Summary Of The Review:**

This paper speculates that contrastive learning is a part-to-whole process and MIM is a part-to-part process and self-supervised learning could learn part-aware representations. It provides an innovative perspective to understand self-supervised learning, supported by comprehensive empirical results and qualitative analyses. The paper is well written but some claims are not well supported. I'm also wondering to what extent the conclusions in this paper can help the future model since the different objectives matter more than the training paradigm (contrastive/MIM). I think this paper is a bit above the borderline.

---

> ### Author Response · Authors · 2022-11-16
> **Response to Reviewer #AQH3 (Part 2/2)**
>
> > **Q2: I do not see a clear reason why iBOT performs the best from the authors' explanation. It's unclear why combining part-to-part and part-to-whole processes could generate a better model. I'm afraid that the differences among the models/objectives would matter more than the differences in the training paradigm (contrastive/MIM). This could make the claims in this paper less useful for guidance on future SSL models.**
>
> Thanks for pointing this out.
> - It is noted that iBOT is built on DINO and has similar training strategies to DINO. Thus, compared with DINO, the main difference of iBOT is that it introduces the MIM training paradigm in its framework to get richer representations. And iBOT authors verified that introducing MIM brings richer semantics to the patch tokens in Sec. 4.3.1 of the iBOT paper. This may be why the combining of MIM and CL brings stronger performance on both object-level and part-level tasks.
> - We agree that the differences among models/objectives would matter though ViT-B is used for all methods. See **our Response to Q1**, we explain the reason the performance of BEiT, MAE, and CAE varies widely across tasks.
> - In this paper, we mainly focus on the **macro comparison** of different pretraining modes (SL, CL, MIM), without studying the micro differences. We propose a measurement of this problem by the **relative performance gap** on part-level tasks and object-level tasks.
>
> For guidance on future SSL models, we summarize several key takeaways below.
>
> 1. Supervised models care more about the whole objects, while self-supervised models have a stronger capability of learning part-aware representations.
> 2. Contrastive learning may learn higher-level semantics and be more semantically abundant than MIM. Combining MIM and CL is potentially capable of learning multi-level semantics.
> 3. MIM methods, *e.g.*, CAE, learn slightly better part-level features than contrastive learning methods. While the learned features of MIM methods, to some extent, depend on the reconstruction targets, *e.g.*, the RGB (color) target used in MAE leads it to learn lower-level features, hence sub-optimal results on high-level tasks.
>
> The takeaways are evidenced correspondingly as follows:
> 1. As shown in **our Response to Q2**, in Fig. 6 and Table 5-6, from object-level to part-level, considerable performance improvements (relative to DeiT) are observed for all self-supervised models. Among these methods, DINO, CAE, and iBOT show larger improvements, demonstrating they can learn better features for part-aware segmentation. For example, DeiT (52.6% in mIoU) outperforms DINO (50.8%) and CAE (47.4%) by 1.8% and 5.2% in ADE20K object-level experiment. However, in ADE20K part-level experiment, the situation is completely opposite. DINO (28.9% in mIoU) outperforms DeiT (27.3%) by 1.6% and CAE (28.4%) outperforms DeiT by 1.1%. Similar results can be seen in segmentation experiments on the Pascal dataset, as well as classification results on COCO and CUB-200.
> 2. As shown in the first row of Fig. 2, the retrieved patches of MoCo v3 are all about dog mouths. While the retrieved patches of CAE contain some patches about the mouths of cats or foxes. Similarly, in Fig. 8, the retrieved patches of CAE for the watch also include patches about the dial of the dial phone. Contrastive learning methods are more likely to retrieve patches with the same category while CAE sometimes retrieves some patches with similar textures. This observation will be more frequently found from the top-100 retrieved patches, based on which we show contrastive learning methods tend to learn better high-level semantics than MIM. And as shown in our paper, iBOT outperforms other methods by large margins in almost all tasks, including object-level and part-level tasks.
> 3. 1\) On part retrieval, CAE outperforms contrastive learning methods by a large margin (about 8% and 5% on CUB-200 and COCO, respectively). On object-level linear probing or segmentation tasks, CAE performs clearly worse than contrastive learning methods. While this performance gap is significantly narrowed even closed on corresponding part-level tasks. 2\) While as shown in **our Response to Q2**, the retrieved patches of MAE in Fig. 7-8 tend to share similar hues, and MAE performs much better than other methods on the human pose estimation task which requires middle-level features.
>
>
> > **Q3: Minor mistakes, in Equations (2) and (3), $P_v$ and $P_m$ are not defined.**
>
> Thank you for pointing it out! We will revise our paper to fix it.
>
> *We sincerely appreciate your comments. Please feel free to let us know if you have further questions.*
>
> Best, \
> Authors

---

> > ### Comment · Reviewer_AQH3 · 2022-11-20
> > **Thanks for the response**
> >
> > Thanks to the authors for dealing with my concerns. Your explanation of the relative performance gap clarifies the claims in the paper, overall, I think it is still reasonable to keep my current score.

---

> ### Author Response · Authors · 2022-11-16
> **Response to Reviewer #AQH3 (Part 1/2)**
>
> We appreciate the positive and insightful comments from you! We address your concerns in detail below.
>
> > **Q1: For the part segmentation part, the authors' explanation, 'possibly due to pretraining quality in representation encoding, BEiT and MAE perform inferior', is not convincing. I do not think it is safe to draw the conclusion based on only one MIM method CAE. Therefore, whether MIM method can be simply described as a part-to-part process is questionable from the experimental results.**
>
> Thanks for pointing it out! Below, we 1) reiterate the **relative performance gap** between part-level tasks and object-level tasks is a main measurement in this paper; 2) explain why BEiT and MAE do not perform better than contrastive learning or supervised learning. We add some new experiments and detailed analysis for MIM methods.
>
> - A very important metric in this work is to measure the relative performance between object-level and part-level, as shown in Fig. 6 and Tab. 5-6. We take DeiT as the baseline to compare its performance with other models in object-level and part-level experiments. In Fig.6 (or see Tab. 5 and 6 for detailed results), from object-level to part-level, the performance (relative to DeiT) of all self-supervised models improves significantly. For example, DeiT (52.6% in mIoU) outperforms MAE (41.0%) by a large margin of 11.6% in the ADE20K object-level experiment. However, in ADE20K part-level experiment, the gap is significantly reduced to 1.0% (27.3% *vs.* 26.3%). Similar results can be seen in Pascal-Object and Pascal-Part experiments. So, although MAE does not outperform DeiT on part-level tasks, we can still find it tends to learn more about parts than DeiT.
>
> - Why BEiT and MAE perform inferior in absolute metrics?
>     - For BEiT, it only obtains 56.7 linear probe on ImageNet while other models achieve about 70 linear probe on ImageNet. The large gap indicates that BEiT does not learn high-quality representations as other models. This could be the main reason that it performs worst on downstream tasks (both object-level and part-level tasks).
>     - For MAE, we find that the colors of the retrieved patches **in Fig. 7 and Fig. 8 in Appendix** are quite similar. Based on it, we speculate that its low-level color reconstruction target helps MAE learn better middle- or low-level features while providing limited high-level semantics. To validate this point, we further conduct experiments on the human pose estimation task, which has been proven by previous work [1] to be a task requiring middle-level features. Similar to linear probing, we train one linear layer over the features extracted by the frozen backbones to predict human keypoints. The results are shown below and MAE outperforms all other counterparts by large margins. So we believe MAE tends to learn lower-level features and pay relatively less attention to the high-level semantics, which leads to its inferior performance on classification and segmentation tasks.
>         | Methods |    AP    |   AP50   |    AR    |   AR50   |
>         | ------- |:--------:|:--------:|:--------:|:--------:|
>         | DeiT    |   6.8    |   27.5   |   14.7   |   44.3   |
>         | MoCo v3 |   12.6   |   42.1   |  21.61   |   56.3   |
>         | DINO    |   16.6   |   49.0   |   25.9   |   62.1   |
>         | MAE     | **23.6** | **59.8** | **33.3** | **69.6** |
>         | CAE     |   17.5   |   50.1   |   28.9   |   64.0   |
>         | iBOT    |   15.5   |   48.1   |   26.0   |   62.8   |
>
> - In summary, the empirical results, analysis, and visualizations validate our claims and show that, generally, MIM and CL can learn part-aware representations. While their performance on different tasks varies due to their special design or properties, e.g., MAE can learn color-aware middle-level representations. We will include more tasks and models in the future.
>
> [1] Ding M, Lian X, Yang L, et al. HR-NAS: Searching Efficient High-Resolution Neural Architectures with Lightweight Transformers. CVPR 2021.

---

### Official Review · Reviewer_nE5Q · 2022-10-24

**Confidence:** 3
**Clarity, Quality, Novelty And Reproducibility:** See above section
**Correctness:** 3
**Technical Novelty And Significance:** 2
**Empirical Novelty And Significance:** 2
**Recommendation:** 5

**Strength And Weaknesses:**

Pros:

The motivation of this paper is clear, the paper is well-written which makes it easy to follow. The experiment setting makes sense to me.

Cons:

The part-to-part and the part-to-whole difference between masked image modeling and contrastive learning is already mentioned in the CAE paper, which makes the finding and conclusion less interesting and surprising.
Some experiment results do not align with the claim well, for example, in the part-retrieval and part-segmentation experiments, BEiT and MAE which are MIM models do not perform better than contrastive learning or supervised learning.
Based on that, no more instructive conclusion about how to better use the property is given (combining two of them like iBOT shouldn't count as a contribution in this paper), which makes the novelty part weak.


**Summary Of The Paper:**

This paper presents an analysis of existing self-supervised learning methods with vision transformers. Extensive experiments are performed on the off-the-shelf pretrained model with the linear and attentive probes to verify the speculation that contrastive learning is a part-to-whole task and masked image modeling is a part-to-part task.

**Summary Of The Review:**

Overall I think this paper goes one step further than CAE and performs an extensive comparison on various benchmarks, which is appreciated. However, some of the experiment results do not align with the authors' claim and the conclusion is somewhat unsurprising and not enlightening. Thus my rating is 5.

---

> ### Author Response · Authors · 2022-11-16
> **Response to Reviewer #nE5Q (Part 2/2)**
>
> > **Q3: Based on that, no more instructive conclusion about how to better use the property is given (combining two of them like iBOT shouldn't count as a contribution in this paper), which makes the novelty part weak.**
>
> According to the joint comparison of our part-level and object-level experiments, we list several key takeaways below, which we believe will inspire future researchers to design better models.
> 1. Supervised models care more about the whole objects, while self-supervised models have a stronger capability of learning part-aware representations.
> 2. Contrastive learning may learn higher-level semantics and be more semantically abundant than MIM. Combining MIM and CL is potentially capable of learning multi-level semantics.
> 3. MIM methods, *e.g.*, CAE, learn slightly better part-level features than contrastive learning methods. While the learned features of MIM methods, to some extent, depend on the reconstruction targets, *e.g.*, the RGB (color) target used in MAE leads it to learn lower-level features, hence sub-optimal results on high-level tasks.
>
> The takeaways are evidenced correspondingly as follows:
> 1. As shown in **our Response to Q2**, in Fig. 6 and Table 5-6, from object-level to part-level, considerable performance improvements (relative to DeiT) are observed for all self-supervised models. Among these methods, DINO, CAE, and iBOT show larger improvements, demonstrating they can learn better features for part-aware segmentation. For example, DeiT (52.6% in mIoU) outperforms DINO (50.8%) and CAE (47.4%) by 1.8% and 5.2% in ADE20K object-level experiment. However, in ADE20K part-level experiment, the situation is completely opposite. DINO (28.9% in mIoU) outperforms DeiT (27.3%) by 1.6% and CAE (28.4%) outperforms DeiT by 1.1%. Similar results can be seen in segmentation experiments on the Pascal dataset, as well as classification results on COCO and CUB-200.
> 2. As shown in the first row of Fig. 2, the retrieved patches of MoCo v3 are all about dog mouths. While the retrieved patches of CAE contain some patches about the mouths of cats or foxes. Similarly, in Fig. 8, the retrieved patches of CAE for the watch also include patches about the dial of the dial phone. Contrastive learning methods are more likely to retrieve patches with the same category while CAE sometimes retrieves some patches with similar textures. This observation will be more frequently found from the top-100 retrieved patches, based on which we show contrastive learning methods tend to learn better high-level semantics than MIM. And as shown in our paper, iBOT outperforms other methods by large margins in almost all tasks, including object-level and part-level tasks.
> 3. 1\) On part retrieval, CAE outperforms contrastive learning methods by a large margin (about 8% and 5% on CUB-200 and COCO, respectively). On object-level linear probing or segmentation tasks, CAE performs clearly worse than contrastive learning methods. While this performance gap is significantly narrowed even closed on corresponding part-level tasks. 2\) While as shown in **our Response to Q2**, the retrieved patches of MAE in Fig. 7-8 tend to share similar hues, and MAE performs much better than other methods on the human pose estimation task which requires middle-level features.
>
>
>
> *Please let us know if you have any further questions for our paper. We really appreciate your time! Thank you!*
>
> Best, \
> Authors

---

> ### Author Response · Authors · 2022-11-16
> **Response to Reviewer #nE5Q (Part 1/2)**
>
> Thank you for your insightful and constructive comments! We have added additional experiments and modified our paper according to your comments.
>
> > **Q1: The part-to-part and the part-to-whole difference between masked image modeling and contrastive learning is already mentioned in the CAE paper, which makes the finding and conclusion less interesting and surprising.**
>
> Thank you for pointing it out! Compared to the CAE paper, our paper provides a deep and systematic analysis with strong evidence. Concretely, we conduct three kinds of quantitative experiments including part retrieval, part classification, and part segmentation on five datasets to support our analysis.
>
> We believe our research goes a step further, making the contrast between different learning paradigms (SL, CL, and MIM), and the specialization of each model clearer.
>
>
> > **Q2: Some experiment results do not align with the claim well, for example, in the part-retrieval and part-segmentation experiments, BEiT and MAE which are MIM models do not perform better than contrastive learning or supervised learning.**
>
> Thanks for pointing it out! Below, we 1) reiterate the **relative performance gap** between part-level tasks and object-level tasks is a main measurement in this paper; 2) explain why BEiT and MAE do not perform better than contrastive learning or supervised learning. We add some new experiments and detailed analysis for MIM methods.
>
> - A very important metric in this work is to measure the relative performance between object-level and part-level, as shown in Fig. 6 and Tab. 5-6. We take DeiT as the baseline to compare its performance with other models in object-level and part-level experiments. In Fig.6 (or see Tab. 5 and 6 for detailed results), from object-level to part-level, the performance (relative to DeiT) of all self-supervised models improves significantly. For example, DeiT (52.6% in mIoU) outperforms MAE (41.0%) by a large margin of 11.6% in the ADE20K object-level experiment. However, in ADE20K part-level experiment, the gap is significantly reduced to 1.0% (27.3% *vs.* 26.3%). Similar results can be seen in Pascal-Object and Pascal-Part experiments. So, although MAE does not outperform DeiT on part-level tasks, we can still find it tends to learn more about parts than DeiT.
>
> - Why BEiT and MAE perform inferior in absolute metrics?
>     - For BEiT, it only obtains 56.7 linear probe on ImageNet while other models achieve about 70 linear probe on ImageNet. The large gap indicates that BEiT does not learn high-quality representations as other models. This could be the main reason that it performs worst on downstream tasks (both object-level and part-level tasks).
>     - For MAE, we find that the colors of the retrieved patches **in Fig. 7 and Fig. 8 in Appendix** are quite similar. Based on it, we speculate that its low-level color reconstruction target helps MAE learn better middle- or low-level features while providing limited high-level semantics. To validate this point, we further conduct experiments on the human pose estimation task, which has been proven by previous work [1] to be a task requiring middle-level features. Similar to linear probing, we train one linear layer over the features extracted by the frozen backbones to predict human keypoints. The results are shown below and MAE outperforms all other counterparts by large margins. So we believe MAE tends to learn lower-level features and pay relatively less attention to the high-level semantics, which leads to its inferior performance on classification and segmentation tasks.
>         | Methods |    AP    |   AP50   |    AR    |   AR50   |
>         | ------- |:--------:|:--------:|:--------:|:--------:|
>         | DeiT    |   6.8    |   27.5   |   14.7   |   44.3   |
>         | MoCo v3 |   12.6   |   42.1   |  21.61   |   56.3   |
>         | DINO    |   16.6   |   49.0   |   25.9   |   62.1   |
>         | MAE     | **23.6** | **59.8** | **33.3** | **69.6** |
>         | CAE     |   17.5   |   50.1   |   28.9   |   64.0   |
>         | iBOT    |   15.5   |   48.1   |   26.0   |   62.8   |
>
> - In summary, the empirical results, analysis, and visualizations validate our claims and show that, generally, MIM and CL can learn part-aware representations. While their performance on different tasks varies due to their special design or properties, e.g., MAE can learn color-aware middle-level representations. We will include more tasks and models in the future.
>
> [1] Ding M, Lian X, Yang L, et al. HR-NAS: Searching Efficient High-Resolution Neural Architectures with Lightweight Transformers. CVPR 2021.

---

> ### Author Response · Authors · 2022-12-03
> **Looking forward to more discussions**
>
> Dear Reviewer nE5Q,
>
> We sincerely appreciate your time and efforts in reviewing our paper, which would help us improve our final paper!
>
> As the deadline for discussion is approaching, we are happy to provide any additional clarifications that you may need. In our previous response, we have carefully studied your comments and made detailed responses summarized below:
>
> - **We highlighted our contributions (against the CAE paper).**
>     - Our paper provides a deep and systematic analysis with strong quantitative evidence for part-aware representation of SSL models.
>     - We believe our research goes a step further, making the contrast between different learning paradigms (SL, CL, and MIM), and the specialization of each model clearer.
>
> - **We detailed our baselines, evaluation metrics, and the reason that BEiT and MAE do not perform well.**
>     - We use the relative performance gap between part-level tasks and object-level tasks to measure the part-aware representation ability.
>     - The pretraining quality of BEiT and the low-level color reconstruction target of MAE limits their performance on high-level part-aware tasks.
>     - To validate the above argument, we add quantitative comparisons on the human pose estimation task which requires middle-level representations.
>
> - **We summarized several key takeaways (instructive conclusions with evidence) for inspiring future works.**
>     - Supervised models care more about the whole objects, while self-supervised models have a stronger capability of learning part-aware representations.
>     - Contrastive learning may learn higher-level semantics than MIM. Combining MIM and CL is potentially capable of learning multi-level semantics.
>     - MIM methods, e.g., CAE, learn slightly better part-level features than contrastive learning methods. While the learned features of MIM methods, to some extent, depend on the reconstruction targets, e.g., the RGB (color) target used in MAE leads it to learn lower-level features, hence sub-optimal results on high-level part-aware tasks.
>
> We hope the above experiments and analyses could clarify your concerns. Please don’t hesitate to let us know if there are any additional clarifications we can offer! Thanks for your time again!
>
> Best, \
> Authors

---

> ### Comment · Reviewer_nE5Q · 2022-12-05
> **Reply to authors**
>
> I appreciate the authors' detailed response. I also checked other reviewers' comments and have similar concerns with them (limited novelty, some words need to be carefully revised to avoid misleading).
>
> Novelty:
>
> Overall I believe this paper is still marginally below the bar of ICLR, as performing existing checkpoints on new datasets/tasks with some analysis and verifying the effectiveness of the existing model (iBOT) is not enough in my perspective.
> Suppose the authors genuinely believe the finding is valuable and non-trivial. In that case, I suggest the author make one step further and propose something new (a model, training/evaluation protocols, etc.) based on the findings, as many other papers did.
>
> Wording:
>
> Although the authors mentioned that we should focus on the relative performance gap between part-level tasks and object-level tasks (which I agree with), words like `BEiT does not learn high-quality representations as other models` (in both discussions and the paper) are very misleading as they can do quite well when fine-tuning on downstream tasks. Whether we should conclude like this (e.g., MAE does not perform well with linear evaluation because it learns lower-level representation from the pixel reconstruction task) is questionable based on the current results.
>
> Overall I believe it is reasonable to keep my rating as borderline reject. I'm also ok if other reviewers and AC believe it meets the conference standard and are willing to accept it.

---

> > ### Author Response · Authors · 2022-12-06
> > **Thanks a lot for your insightful discussion**
> >
> > Thanks for your feedback again, which would help us further improve our paper! Below we provide detailed responses to your remaining two questions about novelty and wording.
> >
> > **For the novelty part**, we respectfully push back on the premise of the question that "performing existing checkpoints on new datasets/tasks with some analysis and verifying the effectiveness of the existing model (iBOT) is not enough".
> > - **Our work is not simply applying existing checkpoints to new tasks/datasets.** Considering part-aware representation is crucial for many downstream tasks, we build a new benchmark containing seven representative models on carefully selected hierarchical datasets, e.g., Pascal, Pascal-Part, ADE20K, and ADE20K-Part. We elaborate our experiments with qualitative analysis, quantitative results, and empirical visualizations. We believe our effort is non-trivial and could inspire future work. We will open-source our benchmarks including all cleaned data, trained models, logs, and visualizations.
> > - **The goal of our work is to understand self-supervised pretraining through studying the capability that self-supervised representation pretraining methods learn part-aware representations.** We provide general explanations and comparisons of the two types of SSL methods (CL and MIM), and show how the properties of each model affect downstream performance, e.g., training strategy, pre-training quality, and reconstruction target.
> > - Verifying the performance of iBOT is not our main focus and it is one of seven representative models in our benchmark. We also show detailed analyses of other models, like MoCO, DINO, MAE, CAE, etc.
> > - This work introduces benchmarks/evaluation protocols of existing models for part-aware tasks. And we agree with you that proposing new models or training protocols based on our findings could make our paper stronger. Thank you for the valuable suggestion and we leave it as our future work.
> >
> > **For the wording part**, we thank the reviewer for the helpful writing suggestions. The claim "BEiT does not learn high-quality representations as other models" should be limited in the linear/attentive prob setting. And the claim "MAE does not perform well with linear evaluation because it learns lower-level representation from the pixel reconstruction task" is validated by our human pose estimation experiments (see our response to Q2) and we will provide more analyses to make it clearer. We will go through our paper and try our best efforts to avoid overclaiming and make our paper more readable.
> >
> >
> > Thank you for your time again! Please don’t hesitate to let us know if there are any additional clarifications we can offer, thanks!
> >
> > Best, \
> > Authors

---

### Official Review · Reviewer_SGtU · 2022-10-24

**Confidence:** 3
**Correctness:** 2
**Technical Novelty And Significance:** 1
**Empirical Novelty And Significance:** 2
**Recommendation:** 6

**Clarity, Quality, Novelty And Reproducibility:**

The paper is clearly written, but the usefulness oft the analysis of Section 3 is to me unclear. The experimental evaluation is exhaustive and in theory reproducible. There seems to be no technical novelty, the paper is a study on SSL/SL and MIM /CL, starting from pretrained models from other papers.


**Strength And Weaknesses:**

### Strengths

S1: the paper is studying part-level recognition, a very interesging and important task

S2: The paper features an extensive empirical study on many object-level recognition and part-level recognition tasks.


### Weaknesses

W1: It is unclear to me what the part-to-whole vs  part-to-part analysis offers, especially with respect to part-aware representations: Sections 3.1 and 3.2 seem to suggest the same thing: Both CL and MIM, "[are] potentially capable of learning part-aware representations.". Are there any insights or interesting experiments that are derived from that analysis and distinction? Cause what i see in Sec 4 is an empirical analysis on which losses can learn part-aware representation, that can stand without section 3, really.

W2: It is unclear to me if the "part-to-whole" effect is only a function of the input (random crops) and the contrastive aspect of the loss; I think that what really matters is what is the contrastive loss appled on: If the loss is on aggregated features from the whole crop, I see this making sense. But what is a contrastive loss is applied at the token level?  How would for example DenseCL (Wang, Xinlong, et al. "Dense contrastive learning for self-supervised visual pre-training." Proceedings of the IEEE/CVF Conference on Computer Vision and Pattern Recognition. 2021.) fits in this framework?

W3: The authors are focusing on random crops, which is only one of the augmentations used for CL - yet there are other augmentations at play. A more fair comparison would use an identical augmentation setup apart from either adding random crops or masking. Are all the other augmentations (color jittering, etc) shared across compared methods?

W4: it is unclear to me, eg from the abstract what is the focus of the study, as I see two  axes that are convoluted: MIM vs CL and SL versus SSL methods. The abstract highlights the first axis, so does Sec 3, but then Figure 2 shows that both CAE (MIM) and MoCov3 (Contrastive) can give part-aware representations, while DeiT (SL) cannot. I think that leaving SL out of the study would make it clearer.

W5: the insights from the empirical study are not analysed enough and can in places seems exargerated:, eg "This implies that in general the self-supervised models are not strong at object-level understanding," this is a very generic statement to make from the results in Table 2.

Q1: Are you retraining all the models under some otherwise identical setup, or using the publicly available models out of the box? Overall, three are many differences between the models compared beyond the loss (from batchsizes, to augmentations, to optimizers to other ViT-related tricks used in different papers).

Q2: Although figure 1 and 3 is mostly figurative it seems to me that it exaggerates the effects of the random crops used in practice. Apart from methods that use local crops as in multi-crop, the scale parameter for the random crops is usually not as large as the one used in this figure. What is the crop scale used in all methods, ie what is the percentage of the max width/height that each readnom crop uses?

Q3: Where are the patches seen in Fig 2 and 4 from? Imagenet?

N1: I wonder how things would change for SL if a more recent variant of the DeiT series was used, e.g. Deit III from
Touvron, Hugo, Matthieu Cord, and Hervé Jégou. "Deit iii: Revenge of the vit." arXiv preprint arXiv:2204.07118 (2022).



**Summary Of The Paper:**

The paper is a study on what is the different when learning with random crops and contrastive learning (CL) in a supervised learning (SL) or self-supervised learning (SSL) way, versus random patch masking via self-supervised learning (SSL) as in the Masked image modeling (MIM) variants (MAE/BeiT/CAE). The authors claim that the former is learning a "part-to-whole" task while the latter a "part-to-part" task; both can lead to strong part-aware representations, while it seems that SL excels more on object-level tasks, while SSL methods like CAE or MoCo-v3 are able to learn part-aware representations.

**Summary Of The Review:**

This paper is at its core an empirical study on part-aware representations; the usefulness of the analysis in section 3 is unclear to me, looking forward for some clarifications from the authors on this. There is no technical novelty, while the insights from the study are also not strong, basically verifying the superiority of iBoT over other SSL methods tested on both object and part tasks. I am looking forward to the author's responses.

--- Post rebuttal update:

I want to thank the authors for constructive discussion. I think iff the authors make the edit they promise to the text  and clarify their contributions, I think I can raise my score to borderline accept. Although limited in novelty, the study in this paper can be valuable to the community, if the contributions are not over-claimed.

---

> ### Author Response · Authors · 2022-11-16
> **Response to Reviewer #SGtU (Part 4/4)**
>
> > **Q1: Are you retraining all the models under some otherwise identical setup, or using the publicly available models out of the box? Overall, there are many differences between the models compared beyond the loss (from batchsizes, to augmentations, to optimizers to other ViT-related tricks used in different papers).**
>
> We use publicly available models to make sure they are well trained. Considering that the training protocols of different models vary widely in most respects (e.g., batch size, augmentation, ViT-related tricks to prevent training instability and crashes), it is unreasonable to use the identical setup. For example, MoCo v3 uses randomly initialized patch embeddings to maintain training stability. Likewise, DINO uses a centering and sharpening strategy to avoid collapse.
>
> In practice, different models may rely on different training strategies to achieve a reasonable performance. And the absolute training quality of the models usually varies. We could measure the **relative performance between object-level and part-level**. Specifically, we could take DeiT as the baseline to compare its performance with other models in object-level and part-level experiments. For example, in Fig. 6 and Tab. 5-6, the gap between DeiT and iBOT on Pascal-Object experiments is marginal (about 0.1% in mIoU). In contrast, when it comes to Pascal-Part experiments, iBOT outperforms DeiT by 3.3% in mIoU.
>
> > **Q2: Although figure 1 and 3 is mostly figurative it seems to me that it exaggerates the effects of the random crops used in practice. Apart from methods that use local crops as in multi-crop, the scale parameter for the random crops is usually not as large as the one used in this figure. What is the crop scale used in all methods, ie what is the percentage of the max width/height that each readnom crop uses?**
>
> We list the crop scale of each method in our experiments, which are the same as the settings in their official code. See below.
> |            |  DeiT  | MoCo v3 |      DINO       |  BEiT  |  MAE  |  CAE   |       iBOT       |
> | ---------- |:------:|:-------:|:---------------:|:------:|:-----:|:------:|:----------------:|
> | Crop Scale | 0.08-1 | 0.08-1  | 0.4-1, 0.05-0.4 | 0.08-1 | 0.2-1 | 0.08-1 | 0.14-1, 0.05-0.4 |
>
> DINO and iBOT use local crops (0.05-0.4), and for other CL methods, the minimal scale is 0.08 and the maximal scale is 1. So the situation is commonly encountered in CL.
>
>
> > **Q3: Where are the patches seen in Fig 2 and 4 from? Imagenet?**
>
> Yes, as stated in **the last paragraph of Appendix A.4**, the patches in Fig. 2 and 4 are from ImageNet. So do Fig. 7 and 8. Concretely, we crop 49 patches from each image and take one patch as the query to search the top 24 similar patches from the validation dataset.
>
> >**N1: I wonder how things would change for SL if a more recent variant of the DeiT series was used, e.g. Deit III from Touvron, Hugo, Matthieu Cord, and Hervé Jégou. "Deit iii: Revenge of the vit." arXiv preprint arXiv:2204.07118 (2022).**
>
> We provide the results on part retrieval and part segmentation using DeiT iii below. It is observed that DeiT iii is still inferior in part-level tasks, which further supports our claims.
>
> |  |  CUB 200 part retrieval (AP) | COCO part retrieval (AP) | ADE20K part seg (mIoU) | Pascal part seg (mIoU) |  LIP part seg (mIoU) |
> | --- |:---:|:---:|:---:|:---:|:---:|
> | DeiT  | 35.0 | 44.1 | 27.3 | 27.4 | 41.4 |
> | **DeiT iii** | 44.1 | 48.9 | 27.8 | 27.6 | 41.9 |
> | MoCo v3 | 50.8 | 52.3| 27.1 | 27.1 | 41.9 |
> | DINO  | 48.9 | 51.8 |28.9 | 27.8 | 41.0 |
> | BEiT | 27.9 | 35.3 |18.6 | 14.8 | 27.2 |
> | MAE | 28.5 | 37.1 | 26.3 | 24.3 | 38.2 |
> | CAE | 58.0 | 57.0 |28.4 | 27.8 | 47.3 |
> | iBOT | 49.3 | 59.2 |32.2 | 30.7 | 44.6 |
>
>
> *We hope that the provided new experiments and additional explanations have convinced you of the merits of our work. Please do not hesitate to contact us if you have other concerns.*
>
> *We appreciate your time! Thank you so much!*
>
> Best, \
> Authors

---

> ### Author Response · Authors · 2022-11-16
> **Response to Reviewer #SGtU (Part 3/4)**
>
> > **W3: The authors are focusing on random crops, which is only one of the augmentations used for CL - yet there are other augmentations at play. A more fair comparison would use an identical augmentation setup apart from either adding random crops or masking. Are all the other augmentations (color jittering, etc) shared across compared methods?**
>
> Thanks for pointing this out. For reference, we list all data augmentations for each method in their official code.
>
> | Augmentations        |   DeiT   | MoCo v3  |   DINO   |   BEiT   |   MAE    |   CAE    |   iBOT   |
> | -------------------- |:--------:|:--------:|:--------:|:--------:|:--------:|:--------:|:--------:|
> | RandomResizedCrop    | &#10004; | &#10004; | &#10004; | &#10004; | &#10004; | &#10004; | &#10004; |
> | RandomHorizontalFlip | &#10004; | &#10004; | &#10004; | &#10004; | &#10004; | &#10004; | &#10004; |
> | RandomGrayscale      | &#10004; | &#10004; | &#10004; |          |          |          | &#10004; |
> | Solarization         | &#10004; | &#10004; | &#10004; |          |          |          | &#10004; |
> | GaussianBlur         | &#10004; | &#10004; | &#10004; |          |          |          | &#10004; |
> | ColorJitter          | &#10004; | &#10004; | &#10004; | &#10004; |          |          | &#10004; |
> | RandomMasking        |          |          |          |          | &#10004; |          |          |
> | BlockwiseMasking     |          |          |          | &#10004; |          | &#10004; | &#10004; |
>
> Different models may rely on different data augmentation strategies to achieve a reasonable performance. It's unreasonable using the same data augmentation strategies for all methods. For example, MAE, which aims to reconstruct the patch color, is more sensitive to color jittering augmentation. So we directly use the official model to conduct our experiments.
>
> For a fair comparison, although the absolute training quality of the models varies, **the relative performance between object-level and part-level is measurable.** For example, in **Fig.6 and Tab. 5-6**, we take DeiT as the baseline to compare its performance with other models in object-level and part-level experiments. We can see that DeiT (92.2% in mIoU) outperforms DINO (88.0%) and CAE (83.3%) by 4.2% and 8.9% in Pascal object-level experiment. However, in Pascal part-level experiment, the situation is completely opposite. Specifically, both DINO and CAE (27.8% in mIoU) outperform DeiT (27.4%) by 0.4%. This indicates that DINO and CAE can learn better features for part-aware segmentation. Similar results can be seen in ADE20K-Object and ADE20K-Part experiments.
>
> > **W4: it is unclear to me, eg from the abstract what is the focus of the study, as I see two axes that are convoluted: MIM vs CL and SL versus SSL methods. The abstract highlights the first axis, so does Sec 3, but then Figure 2 shows that both CAE (MIM) and MoCov3 (Contrastive) can give part-aware representations, while DeiT (SL) cannot. I think that leaving SL out of the study would make it clearer.**
>
> Sorry for the confusion. SL is an important baseline for analyzing SSL models. With SL as a strong baseline, we can measure the relative performance difference between object-level and part-level. As shown in Fig. 6 and Tab. 5-6, we show the superior property of SSL models that they can give part-aware representations.
>
>
> This work also delivers the following messages:
> - SL tends to focus on the whole object.
> - CL learns part and whole representations before and after the projection layer, respectively; and the representation learned by CL may contain higher-level semantics.
> - MIM is naturally designed to learn part representations; The representation learned by MIM is likely to depend on the reconstruction target, *e.g.*, the RGB reconstruction target leads MAE to learn relatively lower-level features.
>
>
>
> > **W5: the insights from the empirical study are not analysed enough and can in places seems exargerated:, eg "This implies that in general the self-supervised models are not strong at object-level understanding," this is a very generic statement to make from the results in Table 2.**
>
> Thanks for pointing it out.
> - Sorry for the confusion caused. Here, we mainly want to say that most self-supervised models are likely to be inferior to supervised model DeiT at object-level understanding, as shown by the results in Table 1-2. We will revise our paper to make it clearer.
> - Table 1-2 only describes the object-level baseline experiments, while this work mainly focuses on part-level experiments. So we analyze the results in Table 1-2 in a concise manner. For more analysis, please refer to **our Response to W1**.

---

> ### Author Response · Authors · 2022-11-16
> **Response to Reviewer #SGtU (Part 2/4)**
>
> For example, in Fig.6, from object-level to part-level, the performance (relative to DeiT) of all self-supervised models improves significantly. Among these methods, DINO, CAE, and iBOT show larger improvements, demonstrating they can learn better features for part-aware segmentation. For example, DeiT (52.6% in mIoU) outperforms DINO (50.8%) and CAE (47.4%) by 1.8% and 5.2% in ADE20K object-level experiment. However, in ADE20K part-level experiment, the situation is completely opposite. DINO (28.9% in mIoU) outperforms DeiT (27.3%) by 1.6% and CAE (28.4%) outperforms DeiT by 1.1%. Similar results can be seen in Pascal-Object and Pascal-Part experiments.
>
>
> > **W2: It is unclear to me if the "part-to-whole" effect is only a function of the input (random crops) and the contrastive aspect of the loss; I think that what really matters is what is the contrastive loss appled on: If the loss is on aggregated features from the whole crop, I see this making sense. But what is a contrastive loss is applied at the token level? How would for example DenseCL (Wang, Xinlong, et al. "Dense contrastive learning for self-supervised visual pre-training." Proceedings of the IEEE/CVF Conference on Computer Vision and Pattern Recognition. 2021.) fits in this framework?**
>
> Thanks for pointing this out!
> - As stated in our paper (*e.g.*, footnote 1), **"we use contrastive learning to refer to the methods that compare random views, e.g., SimCLR, MoCo, and BYOL."** DenseCL [1] is not included in our main focus (CL on view-level features), however, we are willing to add it to our paper and discuss the potential. Considering DenseCL applies contrastive losses on both the view-level features and patch-level features, we expect its patch features tend to be about parts while its class token tends to be object-level.
> - We did a quick experiment with DenseCL. Considering the official checkpoint of DenseCL can not be directly used in our experiments, we have tried to evaluate it on our part-level tasks by migrating its ResNet-50 to ViT-B. Its linear probing performance (44.3%) on ImageNet and LIP segmentation results (19.6% mIoU) were relatively low. We will make further exploration and tune the hyper-parameters (*e.g.*, learning rate) to get proper performance. Once we got the results, we would update them in our paper.
> - Similar to DenseCL, there are also some works [2-4] that introduce patch-level contrastive loss into the MIM paradigm. We will cite and discuss these works in our revision after their codes are published.
>
> [1] Wang X, Zhang R, Shen C, et al. Dense contrastive learning for self-supervised visual pre-training. CVPR 2021. \
> [2] Yi K, Ge Y, Li X, et al. Masked Image Modeling with Denoising Contrast. arXiv preprint arXiv:2205.09616, 2022. \
> [3] Tao C, Zhu X, Huang G, et al. Siamese Image Modeling for Self-Supervised Vision Representation Learning. arXiv preprint arXiv:2206.01204, 2022. \
> [4] Huang Z, Jin X, Lu C, et al. Contrastive masked autoencoders are stronger vision learners. arXiv preprint arXiv:2207.13532, 2022.

---

> ### Author Response · Authors · 2022-11-16
> **Response to Reviewer #SGtU (Part 1/4)**
>
> Thank you for your insightful and constructive comments! We have added additional experiments and will modify our paper according to your comments.
>
> > **W1: It is unclear to me what the part-to-whole vs part-to-part analysis offers, especially with respect to part-aware representations: Sections 3.1 and 3.2 seem to suggest the same thing: Both CL and MIM, "[are] potentially capable of learning part-aware representations.". Are there any insights or interesting experiments that are derived from that analysis and distinction? Cause what i see in Sec 4 is an empirical analysis on which losses can learn part-aware representation, that can stand without section 3, really.**
>
> In this work, we provide **both qualitative analysis (Sec. 3) and empirical results (Sec. 4)** to support our claims. In section 3, we review the representative formula of CL and MIM, and make general explanations of the two types of SSL methods. In section 4, we go a step further and use quantitative metrics to illustrate the properties of each model.
>
> **From the perspective of qualitative analysis**, based on the part-to-whole vs part-to-part analysis in section 3 (also see Fig. 2-5 in the main paper and Fig. 7-8 in Appendix A.5), we speculate that:
> - CL learns part and whole representations before and after the projection layer, respectively;
> - MIM is naturally designed to learn part representations;
> - the MIM reconstruction target may bring target-related properties to the learned representation.
>
>
> **From the perspective of empirical results**, we conduct experiments in section 4 to validate the above assumptions and analyze the characteristics of each model.
> 1. DeiT, as an SL method, cares more about the whole object.
> 2. CL methods, MoCo v3 and DINO, can learn part-aware representations. Also, they may learn higher-level semantics than MIM.
>     - We can find the evidence in the **first row of Fig. 2**, the retrieved patches of MoCo v3 are all about dog mouths. While the retrieved patches of CAE contain some patches about the mouths of cats or foxes. Similarly, observations could be found in Fig. 8. Contrastive learning methods are more likely to retrieve patches with the same category while CAE sometimes retrieves some patches with similar textures.
> 3. CAE shows comparable even better performance than CL on part-aware tasks, though inferior in object-aware tasks.
> 4. The pretraining quality of the BEiT model (56.7% linear probe on ImageNet) is not as good as other models (about 70% linear probe on ImageNet), leading to lower performance in downstream tasks. This indicates that pretraining quality is crucial in absolute performance, thus relative performance between object-level and part-level could be a good metric to show if a model can learn part-aware representations.
> 5. From **Fig. 7 and Fig. 8 in Appendix**, we find that the colors of retrieved patches by MAE are similar to that of the query patch. This is possibly due to its color reconstruction target. The low-level color reconstruction characteristic leads to inferior performance on retrieval and linear/attentive probing tasks which require high-level semantics, but it helps MAE learn better middle- or low-level features. To validate this point, similar to linear probing, we train one linear layer over the features extracted by the frozen backbones to predict human keypoints, which has been proven to be a task requiring middle-level features. The results are shown below and MAE outperforms all its counterparts by large margins.
>     | Methods |    AP    |   AP50   |    AR    |   AR50   |
>     | ------- |:--------:|:--------:|:--------:|:--------:|
>     | DeiT    |   6.8    |   27.5   |   14.7   |   44.3   |
>     | MoCo v3 |   12.6   |   42.1   |   21.6   |   56.3   |
>     | DINO    |   16.6   |   49.0   |   25.9   |   62.1   |
>     | MAE     | **23.6** | **59.8** | **33.3** | **69.6** |
>     | CAE     |   17.5   |   50.1   |   28.9   |   64.0   |
>     | iBOT    |   15.5   |   48.1   |   26.0   |   62.8   |
> 6. iBOT combines CL and MIM so that it is potentially capable of learning stronger semantics and performs the best on many tasks. This could inspire future self-supervised model design.
> 7. In summary, the results, analysis, and visualizations validate our claims and show that, generally, MIM and CL can learn part-aware representations. While their performance on different tasks varies due to their special design or properties, including training strategy, pre-training quality, and reconstruction target.
>
> In this way, a **more important metric** in this work is to measure the **relative performance between object-level and part-level**, as shown in **Fig. 6 and Tab. 5-6**. We take DeiT as the baseline to compare its performance with other models in object-level and part-level experiments.

---

> > ### Comment · Reviewer_SGtU · 2022-11-19
> > **Feedback after authors responses**
> >
> > I want to thank the authors for extensive responses. The new experiments on human keypoint prediction, results using DenseCL, results using Deit-III and the table below which clarifies which augmentations are used in each method are all interesting and  in my opinion should be properly integrated in the paper.
> >
> > This is a study paper and therefore I think it should be clear and not over-claim. There are two main concerns remains for me that do not allow me yet to increase my score:
> >
> > A)  Sec3 is not presented as a "qualitative results" section.
> >
> > The authors say in their response above that
> >
> > > we provide both qualitative analysis (Sec. 3) and empirical results (Sec. 4)
> >
> > however this is not how Sec 3 is presented in the text, i.e. the section title is
> > "**UNDERSTANDING** CONTRASTIVE LEARNING AND MASK IMAGE MODELING"
> >
> > and in the contributions paragraph they claim that
> > > We **explain** masked image modeling as a part-to-part task and contrastive learning as a partto-whole task, and **speculate** that self-supervised pretraining has the potential for learning part-aware representations.
> >
> > I think that a better presentation of that part of the study is exactly as presented in the responses above, i.e. this is the qualitative analysis that is followed by an experimental analysis. I think that the intro and contributions text should reflect that, while also an introductory  paragraph for Sec 3 could clarify that, and say sth like "In this section we perform some qualitative analysis using patch retrieval to  study the capability methods for learning part-aware representations". Also I more fitting title for Sec 3 would be something that mentions "patch retrieval" and "qualitative".
> >
> > B) Usage of the term "CL" throughout the paper:
> > > we use contrastive learning to refer to the methods that compare random views, e.g., SimCLR, MoCo, and BYOL.
> >
> > I strongly disagree with the terminology. "Contrastive learning" extends way more than this very narrow definition, and it is a term that fits pairwise losses regardless of how the pairs are constructed (eg random views) and has been used for many years before those recent advances in SSL. I think that this needs to change for acceptance; authors should replace referring  to "CL" and instead either refer to recent works explicitly by name or collectively as "recent SSL methods that use CL", or think of a more fitting terminology that states the fact that what is compared is random views, e.g.  methods that learn  "by comparing random views" or "via augmentation invariance".

---

> > > ### Author Response · Authors · 2022-11-20
> > > **Response to Your Further Concerns**
> > >
> > > Thanks for your encouraging response. We are glad to see you agree with most of our explanations, and appreciate your detailed and valuable suggestions that will help us improve the quality and clarity of the paper. All the new experiments, analyses, and corrections will be integrated into our paper following your suggestions.
> > >
> > > We provide the following responses to the two newly posed concerns to avoid overclaiming and make our paper more readable.
> > >
> > >
> > > > **Q1: Sec. 3 is not presented as a "qualitative results" section.**
> > >
> > > Sorry for the confusion in Sec. 3. Following your suggestions, we would make the following efforts to make it clearer hence a better-presented contribution.
> > >
> > > - Revised **contribution 2**:
> > >     - We study from two perspectives: 1) a qualitative analysis by patch retrieval to analyze the part-to-part characteristic of masked image modeling and the part-to-whole characteristic of random-view based contrastive learning, and 2) followed experimental results to illustrate the properties of each model.
> > >
> > > - Revised **title of Sec. 3**:
> > >     - Qualitative Analysis of Recent Self-supervised Methods via Patch Retrieval.
> > >
> > > - Add **an introductory paragraph in Sec. 3**:
> > >     - In this section, we qualitatively analyze the capability for learning part-aware representations of recent random-view based contrastive learning methods and MIM methods.
> > >
> > > We will also adapt the introduction section and other contents of Sec.3 accordingly to fit the new modifications and get a better presentation. Thank you!
> > >
> > > > **Q2: Usage of the term "CL" throughout the paper.**
> > >
> > > Thanks for pointing it out. We appreciate the reviewer for the nice and constructive suggestions. We would make the following efforts to avoid overclaiming on the terminology.
> > >
> > > - We replace words that may cause overstatement or misunderstanding, like the term "CL" to "random-view based CL".
> > > - We specify that we are working on recent SSL methods [1-3] that use CL by comparing random views, termed "random-view based contrastive learning".
> > > - We add definitions for "random-view based CL" at the beginning of our paper, and revise all subsequent contents.
> > >
> > > All modifications on language will be delivered in our paper and reflected in the final version as well.
> > >
> > > [1] Chen, Ting, et al. "A simple framework for contrastive learning of visual representations." ICML 2020. \
> > > [2] Chen, Xinlei, Saining Xie, and Kaiming He. "An empirical study of training self-supervised vision transformers." ICCV 2021. \
> > > [3] Grill, Jean-Bastien, et al. "Bootstrap your own latent-a new approach to self-supervised learning." NeurIPS 2020.
> > >
> > > *We sincerely appreciate your time and constructive suggestions which help improve our paper a lot. Hope our responses address all your concerns. Please feel free to let us know if you have further concerns.*
> > >
> > > Best, \
> > > Authors

---

> > > > ### Comment · Reviewer_SGtU · 2022-11-20
> > > > **Thanks again for the response**
> > > >
> > > > I think iff the authors make the edit above and clarify their contributions, I think I can raise my score to borderline accept. Although limited in novelty, the study in this paper can be valuable to the community, if the contributions are not over-claimed. Thanks again for the constructive discussion

---

### Author Response · Authors · 2022-11-16
**General Response: Contributions and New Experiments**

We sincerely appreciate all reviewers’ time and efforts in reviewing our paper. We are glad to find that reviewers generally recognized our contributions:
* **Task.** Very interesting and important task [SGtU]; Clear motivation [nE5Q]; Innovative claim leads people to a more insightful understanding of self-supervised learning [AQH3].
* **Experiments.** Extensive empirical study [SGtU, nE5Q, AQH3]; Experiment setting makes sense [nE5Q]; Both empirical results and qualitative analysis are provided [AQH3].
* **Writing.** Clearly written [SGtU, nE5Q]; Easy to follow [nE5Q]; Clear and intuitive explanation [AQH3].

And we also thank all reviewers for their insightful and constructive suggestions, which help a lot in further improving our paper. In addition to the pointwise responses below, we summarize some takeaways from our part-level and object-level experiments, as well as supporting experiments added in the rebuttal according to reviewers’ suggestions.

**Takeaways**
1. Supervised models care more about the whole objects, while self-supervised models have a stronger capability of learning part-aware representations.
2. Contrastive learning may learn higher-level semantics and be more semantically abundant than MIM. Combining MIM and CL is potentially capable of learning multi-level semantics.
3. MIM methods, *e.g.*, CAE, learn slightly better part-level features than contrastive learning methods. While the learned features of MIM methods, to some extent, depend on the reconstruction targets, *e.g.*, the RGB (color) target used in MAE leads it to learn lower-level features, hence only sub-optimal results on high-level tasks.

**New Experiments**
* Comparison of data augmentations for each method [SGtU].
* Experiments for DeiT iii on five part-level datasets [SGtU]
* Comparisons of all methods on the human pose estimation task which requires middle-level representations [nE5Q, AQH3].

The additional experiments and modifications on language will be delivered in our paper and reflected in the final version as well. We hope our pointwise responses below could clarify all reviewers’ confusion and alleviate all concerns. We thank all reviewers’ time again.

---

### Decision · Program_Chairs · 2023-01-20

**Decision:**

Reject

**Justification For Why Not Higher Score:**

As an analysis paper, the bar in terms of experimental execution is very high. The paper, in its current form does not meet that bar. A lot of constructive feedback was given in the reviews, and the paper can definitely be strengthened accordingly.

**Justification For Why Not Lower Score:**

N/A

**Metareview: Summary, Strengths And Weaknesses:**

This paper presents an interesting study, trying to provide understanding of the differences between MIM and CL self-supervised learning. The draft is not a "method" paper but an "analysis" one, and was rated as such by the committee. Reviewers have praised the importance of the tackled problem, as well as the introduction of a benchmark and experimental protocol for testing part-level understanding.
However, we have also noted that some of contributions are a bit over-claimed and that the analysis presented in the paper could largely be improved. As an analysis paper, the hypothesis should be very well formulated and properly tested. The paper is unclear in many places and some of the conclusions are only supported by qualitative results, which could be interpreted either way. A lot of constructive and actionable feedback was provided in the discussion phase; I believe that if the authors would selected the studied methods better, added domains, re-written and strengthened the manuscript, this could be a really interesting study. In its current form, I recommend the paper for rejection, and encourage the authors to improve the manuscript and submit to another venue.

**Summary Of Ac-Reviewer Meeting:**

This paper has received consistent scores in the borderline range. During the meeting, we have discussed :
- The fact that the paper copes with an interesting problem.
- That this is an analysis paper, not a method one. We agreed that this should not penalise the paper by itself, but definitely raises the bar in terms of expectations for the evaluations.
- We agreed that execution of the analysis could be improved, and that the paper is not ready for publication.